# Active and conductive layer stacked superlattices for highly selective CO$_2$ electroreduction

Junyuan Duan[1], Tianyang Liu[2], Yinghe Zhao [1], Ruoou Yang[1], Yang Zhao[1], Wenbin Wang[1], Youwen Liu [1✉], Huiqiao Li [1], Yafei Li [2✉] & Tianyou Zhai [1✉]

Metal oxides are archetypal CO$_2$ reduction reaction electrocatalysts, yet inevitable self-reduction will enhance competitive hydrogen evolution and lower the CO$_2$ electroreduction selectivity. Herein, we propose a tangible superlattice model of alternating metal oxides and selenide sublayers in which electrons are rapidly exported through the conductive metal selenide layer to protect the active oxide layer from self-reduction. Taking BiCuSeO super-lattices as a proof-of-concept, a comprehensive characterization reveals that the active [Bi$_2$O$_2$]$^{2+}$ sublayers retain oxidation states rather than their self-reduced Bi metal during CO$_2$ electroreduction because of the rapid electron transfer through the conductive [Cu$_2$Se$_2$]$^{2-}$ sublayer. Theoretical calculations uncover the high activity over [Bi$_2$O$_2$]$^{2+}$ sublayers due to the overlaps between the Bi $p$ orbitals and O $p$ orbitals in the OCHO* intermediate, thus achieving over 90% formate selectivity in a wide potential range from −0.4 to −1.1 V. This work broadens the studying and improving of the CO$_2$ electroreduction properties of metal oxide systems.

[1] State Key Laboratory of Materials Processing and Die & Mould Technology, and School of Materials Science and Engineering, Huazhong University of Science and Technology, 430074 Wuhan, Hubei, P. R. China. [2] Jiangsu Collaborative Innovation Centre of Biomedical Functional Materials, Jiangsu Key Laboratory of New Power Batteries, School of Chemistry and Materials Science, Nanjing Normal University, 210023 Nanjing, Jiangsu, P. R. China. ✉email: ywliu@hust.edu.cn; liyafei@njnu.edu.cn; zhaity@hust.edu.cn

Electrochemical $CO_2$ reduction ($CO_2RR$) to high value-added carbon-based products has received extensive interest and represents a path from renewable electricity sources to chemical and fuel production[1–4]. Formate, which is widely used as the most kinetically accessible feedstock in the fields of pharmaceuticals, textiles, and energy, is an attractive liquid product of $CO_2RR$[5–7]. Metal oxides (such as $SnO_2$, $Bi_2O_3$, and $In_2O_3$) as the most common and widely available catalysts have been extensively explored to achieve $CO_2$ electroreduction to formate with considerable activity[5,8–12]. The development of advanced in situ characterization has revealed that metal oxides can undergo in situ self-reduction to zero-valence metal during the $CO_2RR$ (Fig. 1a)[13,14]. Substantial efforts have been devoted to exploring efficient catalysts derived from these metal oxides[15–19]. However, with this self-reduction, the competitive hydrogen evolution reaction (HER) performance of the derived metal catalysts will gradually dominate[1,14,19–23], resulting in their $CO_2RR$ activity being difficult to maintain in a wide potential window. Actually, this spontaneous self-reduction of metal oxides is contradictory to the high selective $CO_2RR$ performance. More importantly, due to the untenable oxidation state during the $CO_2RR$, unraveling the catalytic mechanism of metal oxides will be inaccessible. Therefore, developing electrocatalysts based on metallic oxidation states rather than on their self-reduction is highly desirable for performance optimization and mechanism exploration, which remains an enormous challenge.

Recently, several representative reports have presented metal oxide–support interaction systems to stabilize the oxidation state during the $CO_2RR$ (Fig. 1a)[15,24–27], especially metal oxide–carbon[15,18,24,26–28], in which the carbon layer acts as a conductive layer to quickly conduct electrons and prevent the active oxides from being reduced. Notwithstanding these efforts, the composite catalytic materials have their own inevitable weaknesses, mainly poor interface contact. As a result, a large

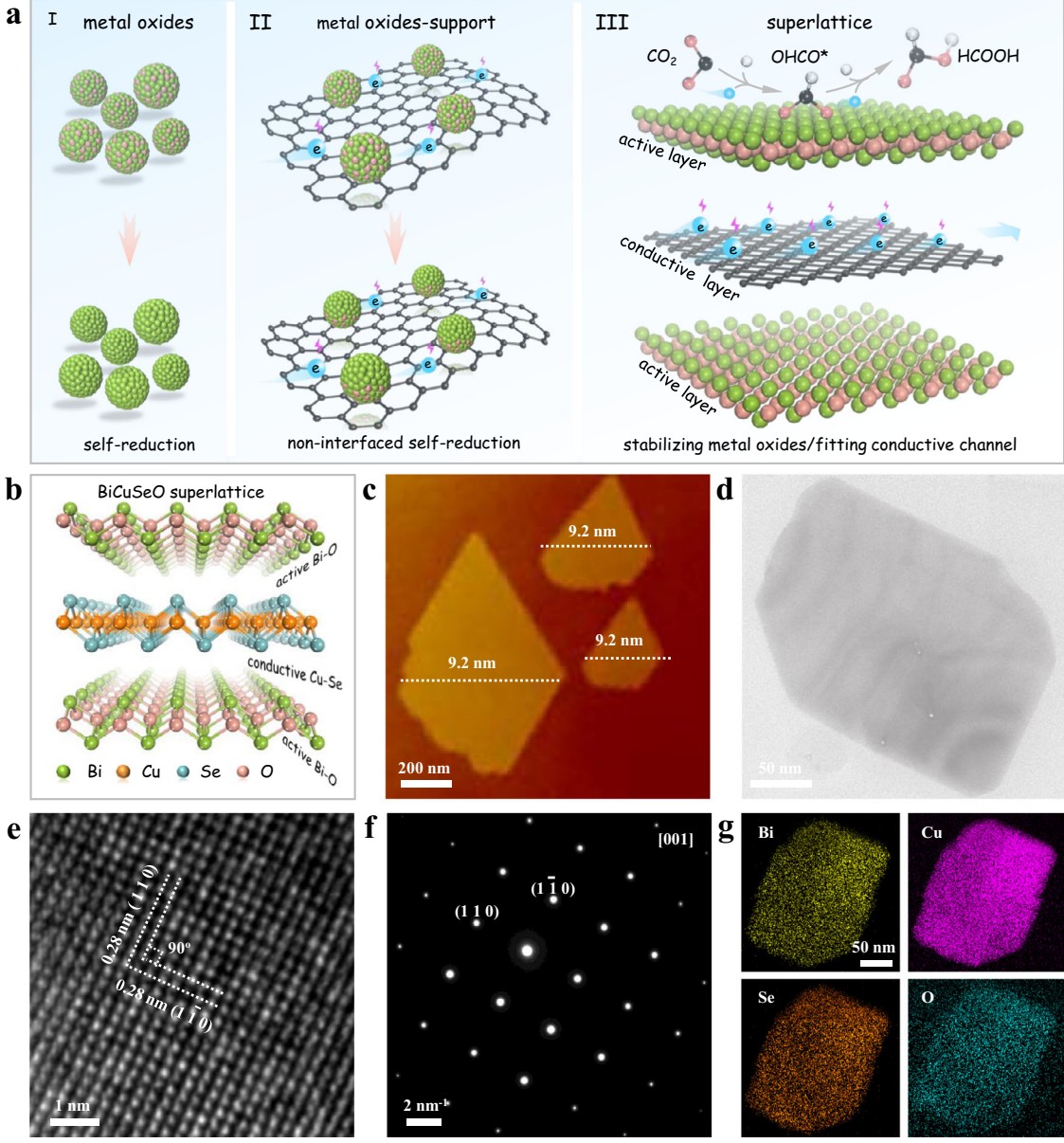

**Fig. 1 Design scheme and structural characterizations of BiCuSeO superlattice nanosheets. a** Schematic illustration of exploring single-phase materials with active/conductive layer alternately stacked superlattices. (I) Pure metal oxides. (II) Metal oxides–support interaction system. (III) Superlattice consisting of a one-by-one vertically stacked active layer (metal oxides) and conductive layer subunit. **b** Structural model of BiCuSeO superlattice. **c** AFM image. **d** TEM image. **e** HRTEM image. **f** Corresponding SAED pattern. **g** EDS mapping images.

number of oxides in noninterfacial areas may still be self-reduced[9,28]. Inspired by the strengths and weaknesses of this structural model, finely stacking an ultrathin oxide layer upon layer on conductive substrates will maximize the potential of such models but require an extremely rigorous synthesis method.

Focusing on this vision, natural superlattice, which consists of one-by-one vertically stacked plural subunits, will be presented as perfect and tangible models. In a natural superlattice, one can obtain a metal oxide layer and conductive layer simultaneously by elaborately designing the prototype of each subunit (Fig. 1a), in which the conductive layer rapidly conducts electrons to protect the active center of the oxide layer to drive the activation of $CO_2$. Guided by this delicate avenue, a typical natural superlattice, BiCuSeO oxyselenides (Fig. 1b), which consists of conductive $[Cu_2Se_2]^{2-}$ and insulating $[Bi_2O_2]^{2+}$ sublayers stacking alternately along the $c$-axis[29–31], is thought to interplay with the Cu–Se layer/Bi–O layer for efficient $CO_2$ electroreduction. More characteristically, Bi-based oxides are the first self-reduction phenomenon discovered and are still considered to be highly promising candidates for selective formate production[18,28,32–34]. Therefore, a natural BiCuSeO superlattice provides an excellent opportunity to obtain high-efficiency $CO_2$ properties in the metal oxidation state rather than their self-reduction, and to explore their structure-activity relationships

Herein, we propose BiCuSeO nanosheets (Ns) as an example and utilize alternately stacked insulating $[Bi_2O_2]^{2+}$ and conductive $[Cu_2Se_2]^{2-}$ in their superlattices as active/conductive sublayers to stabilize the metal oxidation state for high $CO_2RR$ activity and selectivity. Specifically, X-ray photoelectron spectroscopy (XPS), synchrotron-based X-ray absorption near-edge structure (XANES) spectroscopy, and extended X-ray absorption fine structure (EXAFS) studies consistently confirm that the Cu–Se sublayers in the BiCuSeO superlattices mainly conduct electrons so that the highly active Bi in the Bi–O layer still holds its oxidation state fine rather than self-reduced zero-valence Bi metal during the $CO_2RR$. Furthermore, density functional theory (DFT) simulation reveals that Bi–O coordination in $[Bi_2O_2]^{2+}$ exhibits a strong coupling effect with its Bi $p$ orbitals overlapping with O $p$ orbitals in OCHO* and enables a highly selective $CO_2RR$ to formate. Benefitting from the interplay of the Cu–Se layer/Bi–O layer, natural BiCuSeO superlattices exhibit a high catalytic selectivity featuring a formate Faradaic efficiency FE of >90% over a wide potential range from −0.4 to −1.1 V. Importantly, the catalytic model, active/conductive layer alternately stacked natural superlattices, could provide valuable insights for the development of highly efficient $CO_2RR$ catalysts and beyond.

## Results

### Structural characterizations of BiCuSeO Ns.
Herein, ultrathin BiCuSeO nanosheets are synthesized by using poly-vinylpyrrolidone (PVP) as the control agent via a mild hydrothermal route. The X-ray diffraction pattern (XRD), Raman spectrum, and calculated formation energy of $[B_2O_2]^{2+}$ are shown in Supplementary Fig. S1, all solidly indexed to the successful synthesis of the crystalline BiCuSeO phase with high purity and stability. The atomic force microscopy (AFM) and transmission electron microscopy (TEM) characterizations in Fig. 1c, d reveal sheet-like morphology with an average thickness of ~9.2 nm of obtained BiCuSeO. The high-resolution TEM (HRTEM) image shows two sets of mutually perpendicular lattice fringes with a spacing of ~0.28 nm (Fig. 1e), corresponding to the (110) and (1-10) planes of tetragonal BiCuSeO. A single set of diffraction spots with fourfold symmetry in the selected area electron diffraction (SAED) pattern (Fig. 1f) illustrates the high orientation along the $c$-axis of the as-obtained BiCuSeO Ns,

which is consistent with HRTEM analysis. Additionally, the energy dispersive spectroscopy (EDS) mapping analysis points out the uniform distribution of Bi, Cu, Se, and O elements (Fig. 1g). Consequently, all of the above structural characterization results evidently verify the successful synthesis of ultrathin BiCuSeO single-crystal nanosheets.

### Electrochemical performances.
The electrochemical performance of the as-fabricated BiCuSeO is evaluated using a three-electrode flow cell in a $CO_2$-saturated $KHCO_3$ aqueous solution (Supplementary Fig. S2). To investigate the effect of Bi–O and Cu–Se sublayers in BiCuSeO for the $CO_2RR$, $Bi_2O_3$, $Cu_2Se$ Ns, and $Cu_2Se$-$Bi_2O_3$ heterostructures (CuSe-BiO) are prepared, and their structural characterizations are displayed in Supplementary Fig. S3, S4 and S5. The corresponding electrocatalytic activities are also tested for comparisons, as shown in Fig. 2. First, the linear sweep voltammetry (LSV) curves are conducted in a potential range of 0 to −1.1 V (the reversible hydrogen electrode, RHE, all potentials mentioned in the following are RHE). As shown in Fig. 2a, BiCuSeO reveals significantly improved current density and more positive onset potential than $Cu_2Se$, $Bi_2O_3$, and CuSe-BiO, demonstrating its better electrocatalytic performance for the $CO_2RR$. To identify the products and their Faraday efficiency (FE) at different potentials, we conducted the electrolysis at a variety of constant potentials from −0.4 to −1.1 V and collected samples for further test. From the chronoamperometry curves (Fig. 2b), the current density is consistent with the LSV curves and remains steady, suggesting the good electrochemical stability of BiCuSeO catalysts. Accordingly, the gaseous and liquid products were quantitatively analyzed by online gas chromatography (GC) and $^1H$ nuclear magnetic resonance ($^1H$ NMR) spectroscopy, respectively. The GC and NMR results show that formate is the predominant product, accompanied by minor amounts of $H_2$ and CO gas. Apparently, the BiCuSeO shows high selectivity toward formate production, and its FE ($FE_{formate}$) is over 90% in a wide potential window ranging from −0.4 to −1.1 V, with a maximum value that can reach ~93.4% at −0.9 V, while the FE for CO and $H_2$ gas are ~2.4% and ~2.4%, respectively (Fig. 2c). Noticeably, the overpotential for formate generation is as low as 190 mV (Supplementary Fig. S6), which is smaller than that of most other Bi-based catalysts[10,26,27,32–43]. Moreover, the Faraday efficiency of formate ($FE_{formate}$) for BiCuSeO is much higher than that of $Cu_2Se$ (the maximum $FE_{formate}$ is ~60%), $Bi_2O_3$ (the maximum $FE_{formate}$ is ~85%), and CuSe-BiO (the maximum $FE_{formate}$ is ~91.7%) at the tested potentials, indicating that BiCuSeO Ns is more inclined to yield the formate product (Fig. 2d, Supplementary Table S1). Furthermore, the calculated formate partial current densities ($J_{formate}$) of BiCuSeO are significantly larger than those of $Cu_2Se$, $Bi_2O_3$, and CuSe-BiO at the same potentials, and the maximum value can reach ~47.5 mA cm$^{-2}$ at −1.1 V (Fig. 2e). These results underlie that BiCuSeO can hold outstanding formate selectivity over a wide potential window (>90% from −0.4 to −1.1 V, Fig. 2f, Supplementary Table S2) and precede other state-of-the-art Bi-based catalysts[10,26,27,32–43]. Moreover, BiCuSeO exhibits a significantly larger current density (a maximum of ~219 mA cm$^{-2}$) and similar good formate selectivity (over 90%) over a wide potential window with iR compensation to eliminate the interference of solution resistance (Supplementary Fig. S7). In general, the $CO_2$ reduction rate will increase obviously in the alkaline electrolyte. Therefore, the electrocatalytic performance of $CO_2$ was also tested in 1 M KOH and exhibited similar outstanding formate selectivity over a wide potential window and an impressive current density ~267 mA cm$^{-2}$ at −1.1 V (Supplementary Fig. S8). Interestingly, the BiCuSeO exhibits the largest

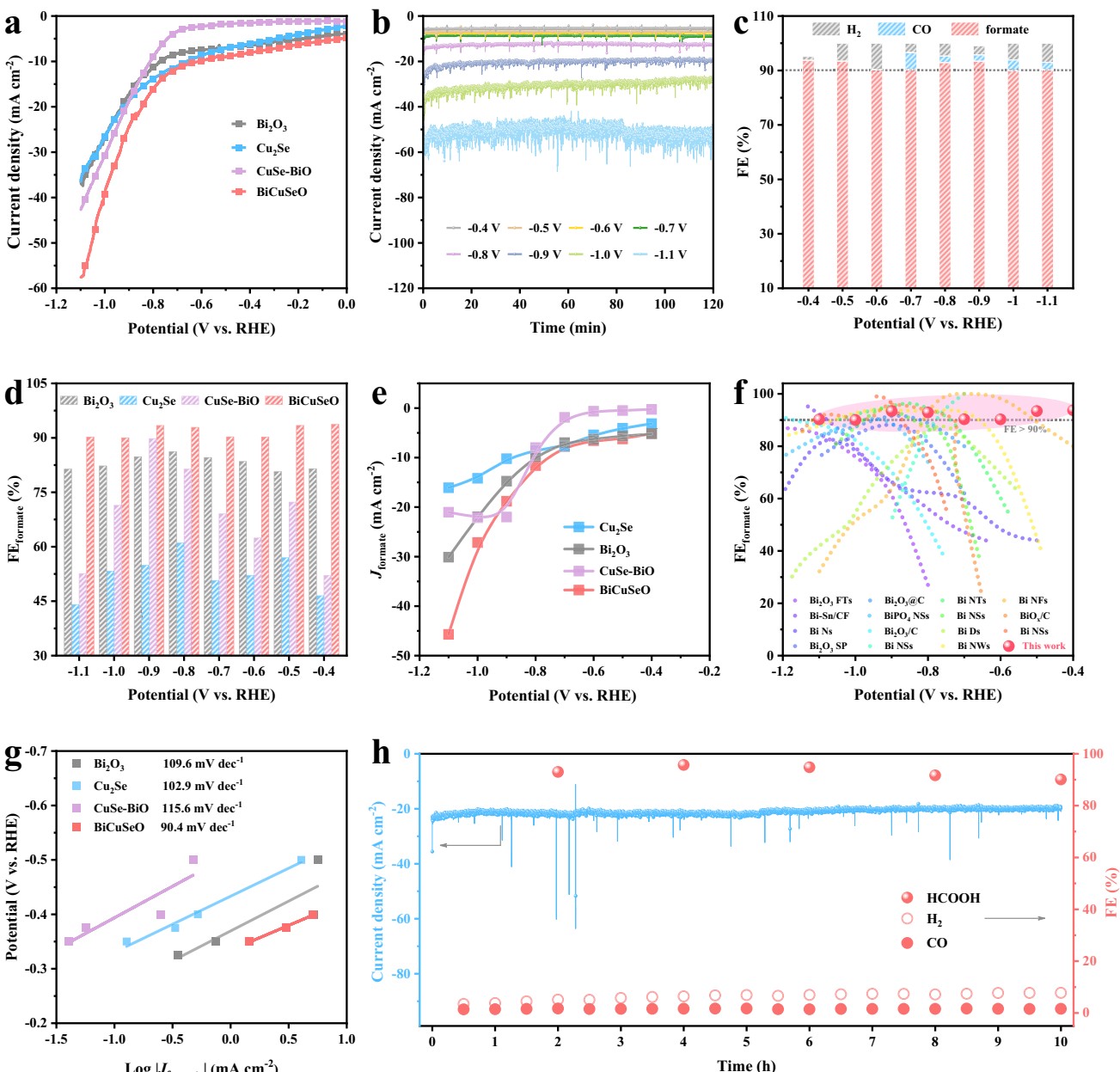

**Fig. 2 CO₂RR performance in 0.5 M KHCO₃. a** LSV curves of BiCuSeO, Cu₂Se, Bi₂O₃ Ns, and CuSe-BiO without iR compensation. **b** Chronoamperometric curves at different potentials. **c** FE of formate, CO and H₂ for BiCuSeO Ns. **d** $FE_{formate}$, **e** $J_{formate}$, and **g** Tafel slope comparisons of BiCuSeO, Cu₂Se, Bi₂O₃ Ns, and CuSe-BiO. **f** $FE_{formate}$ comparison among the reported Bi-based catalysts and the BiCuSeO[10, 26, 27, 32–43]. **h** 10 h Chronoamperometry results for BiCuSeO at −0.9 V.

electrochemical active surface area (ECSA) and ECSA-normalized formate current densities, suggesting a large increase in intrinsic activities refer to Bi₂O₃, Cu₂Se, and CuSe-BiO (Supplementary Fig. S9, S10 and Table S3). In addition, the Tafel slope is determined by using the logarithm of formate partial current density against the applied potentials to evaluate the reaction kinetics of CO₂ RR[44]. The calculated Tafel slope of BiCuSeO is ~90.3 mV dec⁻¹, which is smaller than that of Cu₂Se (~102.9 mV dec⁻¹), Bi₂O₃ (~109.6 mV dec⁻¹), and CuSe-BiO (~115.6 mV dec⁻¹), suggesting the favorable kinetics of BiCuSeO for formate generation (Fig. 2g). Furthermore, the electrochemical reduction reaction for CO₂ at a fixed potential of −0.9 V is carried out over an extended period to evaluate the stability of the BiCuSeO. As shown in Fig. 2h, the total current density stabilizes at ~21 mA cm⁻² together with an average $FE_{formate}$ of ~94% over 10 h. Remarkably, by the SEM images, the morphology

of BiCuSeO catalysts is substantially preserved (Supplementary Fig. S11). In summary, all the above results indicate that the BiCuSeO exhibits efficient formate selectivity in a wide potential window for the CO₂RR.

**Intermediates detection**. To monitor the reaction process and the intermediate species of CO₂RR, we carried out in situ electro-chemical Raman spectroscopy tests (Fig. 3a). Two obvious Raman peaks at 1160 and 1540 cm⁻¹ are detected at −0.5 V or lower potentials in Fig. 3b, c. To identify the ascription of these vibration peaks, the Raman spectra of possible intermediate groups are calculated and shown in Fig. S12. Specifically, the peaks at 1160 cm⁻¹ can be ascribed to the C=O stretching vibration of surface-adsorbed carbonate ($\nu_s CO_2 \cdot^-$) during the CO₂RR electrolysis process

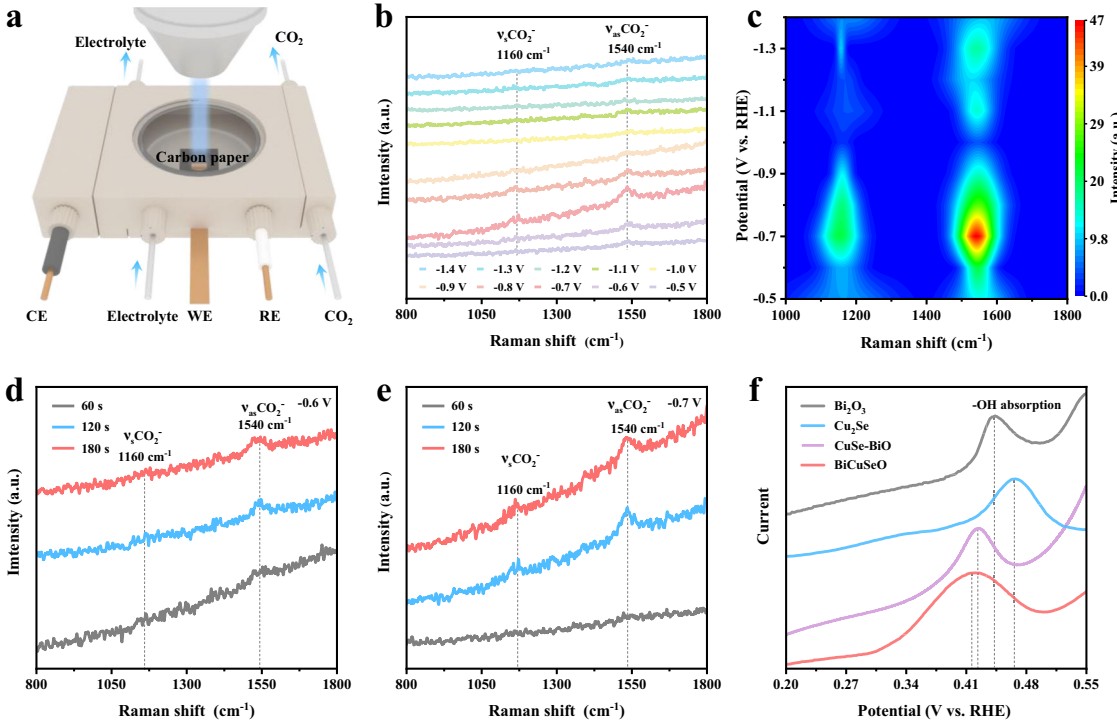

**Fig. 3 Intermediates detection on BiCuSeO during CO₂RR. a** Schematic illustration of the in situ Raman measurement device during the CO₂RR.
**b**, **c** Potential- dependent and **d**, **e** time-dependent in situ Raman spectra of BiCuSeO catalysts in 0.5 M KHCO₃ solution under CO₂ bubbling. **f** Single
oxidative LSV scans in N₂-saturated 0.1 M NaOH solution for BiCuSeO, Cu₂Se, Bi₂O₃ Ns, and CuSe-BiO.

(Supplementary Fig. S12a, Table S4)[45–47]. Meanwhile, the peaks
located at 1540 cm⁻¹ are vibrational fingerprints of asymmetric
C–O stretching vibration modes of proton-trapped carboxylate
*CO₂·⁻ radicals (OCHO*, Supplementary Fig. S12b, Table S5)[45–47].
Therefore, both peaks are attributed to the key intermediates
OCHO* for the formate product during electroreduction CO₂[46–48],
which also confirms that CO₂ could be directly activated and
reduced into formate with lower potential on BiCuSeO catalysts
(Supplementary Fig. S13). Notably, their peak intensity gradually
increases as the applied potentials and reaches a high point at
−0.7 V. This variation trend of characteristic peak intensity with
potential may depend on the trade-off between adsorption and
transformation of intermediate products. Moreover, the time-count
in situ Raman spectra at −0.6 and −0.7 V show that the peaks
intensities of the crucial intermediates for formate gradually
strengthen with expanding time (Fig. 3d, e), resulting from
favorable adsorption and proton-trapping capacity of formate
intermediates. This indicates a step-by-step reaction process for
formate generation from CO₂ (Supplementary Fig. S13). No obvious
band associated with C–O stretching or C=O stretching of the
*COOH intermediate for the CO product indicates that the
formation of CO on BiCuSeO is almost suppressed[45–47,49], which is
consistent with the above experimental FE results. To validate the
binding affinity of CO₂·⁻, the adsorption of OH⁻ as a surrogate for
CO₂·⁻ is detected by oxidative LSV scans in a N₂-bubbled 0.1 M
NaOH electrolyte. Figure 3f reveals that the potential for surface
OH⁻ adsorption on BiCuSeO is lower than that for Bi₂O₃, Cu₂Se Ns,
and CuSe-BiO. This result combined with the Raman analyses
adequately illustrates that the BiCuSeO possesses a stronger
adsorption affinity for OH⁻, and hence they could efficiently
stabilize the CO₂·⁻ intermediate, finally facilitating formate
production.

**Structural transformation, XAFS, XPS, and TEM characterizations of BiCuSeO after CO₂RR.** Metal oxides-based

electrocatalysts inevitably undergo spontaneous reduction under
the function of negative potential during the CO₂RR process. The
XRD pattern and Raman spectrum in Supplementary Fig. S14
show that the crystalline BiCuSeO phase can still be mainly
retained with partial Se escaping during CO₂RR process. To
further determine whether the structure of the Bi–O layer can be
maintained as expected, the synchrotron radiation X-ray
absorption fine structure (XAFS) spectroscopy was performed.
Specifically, Bi L3-edge XAFS measurements are explored and
presented in Fig. 4a. Both the absorption edge and white line peak
of the XANES for BiCuSeO nearly overlap with that of the Bi₂O₃
reference, suggesting the Bi³⁺ species of Bi atoms in BiCuSeO.
After the CO₂RR, the absorption edge of BiCuSeO_R (the BiCuSeO
after the CO₂RR is denoted as BiCuSeO_R) only slightly shifts
toward lower energy, indicating that the oxidation state of Bi is
mainly retained. Furthermore, the Fourier transform (FT) of the
EXAFS curve is resolved to evaluate the Bi local environment at
the atomic level (Fig. 4b). The intense peak of BiCuSeO_R is
located at 1.63 Å (without chemical shift), which is still consistent
with the pristine BiCuSeO and can be attributed to the first Bi–O
coordination shell. The relatively controlled peak weakening
further explains why most of the Bi–O bonds of BiCuSeO are
maintained after the CO₂RR. Meanwhile, wavelet transform
(WT) is used to precisely analyze the Bi L3-edge extended X-ray
absorption fine structure (EXAFS) oscillations. The WT contour
plots of BiCuSeO_R display only one intensity maximum at
~1.6 Å, which apparently corresponds to the Bi–O coordination
with BiCuSeO rather than the Bi–Bi in Bi powders. Notably,
combined with its crystal structure, the length of the Bi–O bond
in the BiCuSeO superlattice is significantly larger than that in the
common Bi₂O₃, showing a specific type of Bi–O coordination
structure for the CO₂RR. Furthermore, to clearly illustrate the
coordination states, the intense peak is finely fitted and the fitting
result suggests that it consists of two superimposed peaks, named
Bi–O₁ (2.13 Å) and Bi–O₂ (2.27 Å) (Supplementary Figs. S15–18,

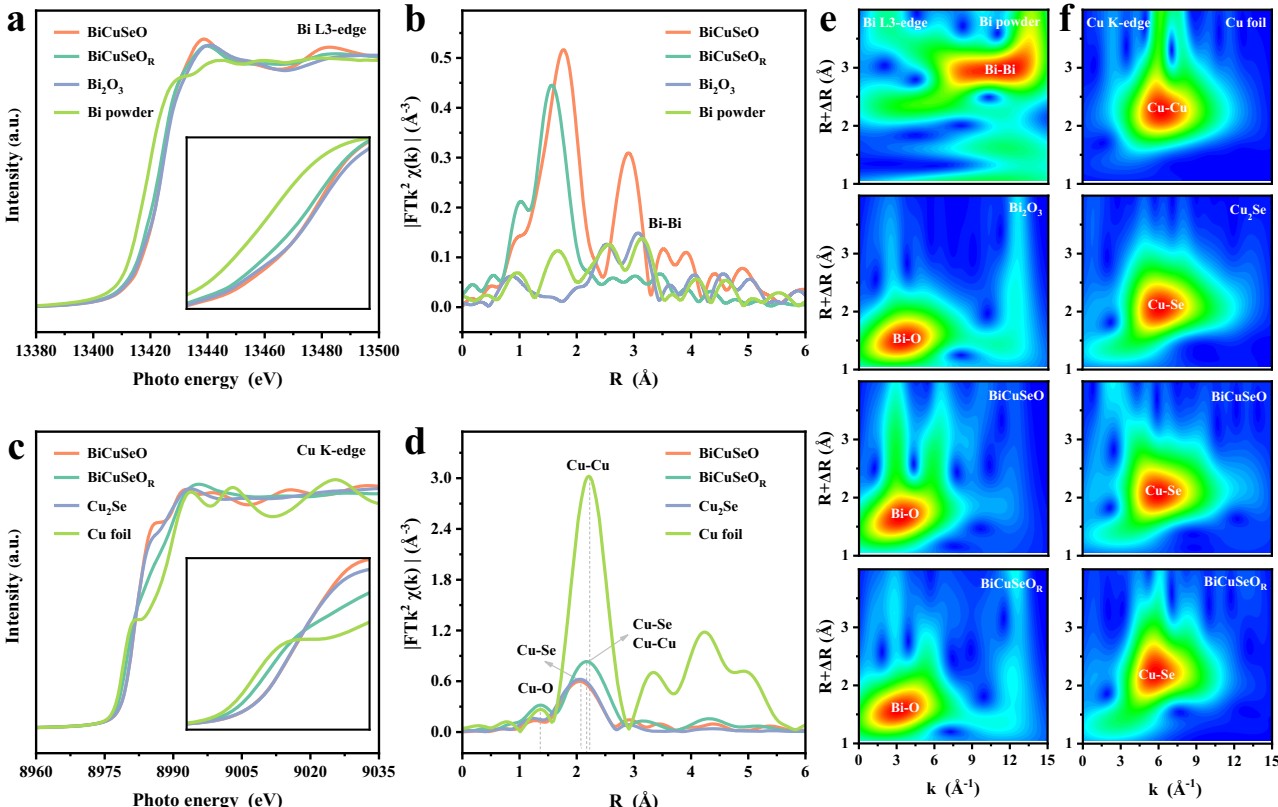

**Fig. 4 XAFS characterizations of BiCuSeO after CO₂RR. a** Bi L3-edge XANES and **b** corresponding FT-EXAFS spectra of BiCuSeO after CO₂RR. **c** Cu K-edge XANES spectra and **d** corresponding FT-EXAFS spectra of BiCuSeO after CO₂RR. Wavelet transform of the $k^2$-weighted EXAFS data of **e** Bi L3-edge and **f** Cu K-edge.

Table S6). The Bi–$O_2$ is attributed to the lattice bond in BiCuSeO, and the coordination number (CN) is calculated to be 2.5 (Supplementary Table S6), which is close to the value for BiCuSeO (CN = 4), further illustrating that only a small amount of reduced $Bi^{3+}$ after the CO₂RR is different from contrastive $Bi_2O_3$ Ns. The fitted coordination number of the metal Bi–Bi bond (3.09 Å) for BiCuSeO$_R$ is only 0.3 (Supplementary Table S6), which also supports the above conclusion. Combined with the in situ Raman results, the appearance of Bi–$O_1$ can be reasonably attributed to chemisorbed *OH and OCHO* species, suggesting the effective adsorption of intermediates in CO₂RR. Taken together, all these results clearly illustrate that Bi sites are still retained in the oxidation state under the negative potential during the CO₂RR, as assumed, which certainly is conducive to driving the conversion of CO₂ to formate (Fig. 4a, b, e, Supplementary Figs. S14–19, Table S6). In contrast, combined with XAFS, XRD, and XPS, $Bi_2O_3$ and CuSe-BiO do essentially transform the metal Bi as reported (Fig. 4a, b, e, Supplementary Figs. S19, S20).

In addition to the Bi–O bond, it is noteworthy that the Bi–Se bond weakens dramatically, with the coordination number reduced from 8 to 0.8 (Fig. 4b, Supplementary Table S6), which results from the precipitation of Se. In general, metal selenides are very sensitive and may undergo structural evolution to form metal oxide catalysts during the electrochemical process. Meanwhile, the Cu–Se sublayer as a conductive layer undertakes electron transport, leading to self-reduction. Therefore, the local structure of Cu–Se sublayers is also investigated by Cu K-edge XAFS spectra (Fig. 4c). First, the absorption edge for the Cu K-edge XANES spectrum of BiCuSeO$_R$ is located between the Cu foil and BiCuSeO, implying that $Cu^+$ species are partially reduced and the valence state of its Cu atoms is between 0 and +1. Furthermore, the EXAFS curves of the Cu K-edge reveal that the

main peak at 2.06 Å disappeared and two new characteristic peaks appeared at 1.93 Å and 2.58 Å for BiCuSeO$_R$ (Fig. 4d, Supplementary Figs. S15, S17, S21, Table S6), respectively. Based on the comparison, the emergent peaks can be attributed to Cu–O bonds and metallic Cu–Cu bonds. Meanwhile, the calculated coordination numbers for Cu–Se bonds, Cu–O bonds, and metallic Cu–Cu bonds are 0.7, 0.8, and 4, respectively (Supplementary Table S6). This result indicates ~17.5 atom% Se atoms are retained in BiCuSeO$_R$. Furthermore, inductively coupled plasma mass spectrometry (ICP-MS) shows that ~11 atom% Se are reserved in BiCuSeO$_R$, which is consistent with XAFS result. Based on the above structural characteristics, it is reasonable to assume that in the process of CO₂ reduction, a large number of Se atoms overflow from the conductive Cu–Se lattice. To keep the structure stable, some of the space left was filled with ambient oxygen, while some of the copper atoms bonded directly (Fig. 5a). The detailed structure transformation process of BiCuSeO is explained in Supplementary Note 1. Similarly, the in situ XANES, EXAFS, and XPS (Figs. 4f, 5a, Supplementary Figs. S21, S22, S23, and Table S6) conformally point to their structural features and further reinforce our inference. As a control, we also explore the structural evolution process of $Cu_2Se$ and CuSe-BiO during CO₂ reduction. Similar to other conventional selenides and Cu–Se layers of BiCuSeO, Se atoms essentially escape from the $Cu_2Se$ lattice and thus completely transform into Cu and $Cu_2O$ (Supplementary Figs. S20, S24, S25).

To further confirm the structural features, X-ray photoelectron spectroscopy (XPS) is also carried out. From Fig. 5b, the intensity of two typical Se 3d characteristic peaks exhibits a sharp decrease for BiCuSeO$_R$ compared to that for BiCuSeO[30,50], suggesting that a large percentage of Se escapes from the Cu–Se sublayers during the electrocatalytic reaction. Meanwhile, the indistinct Se

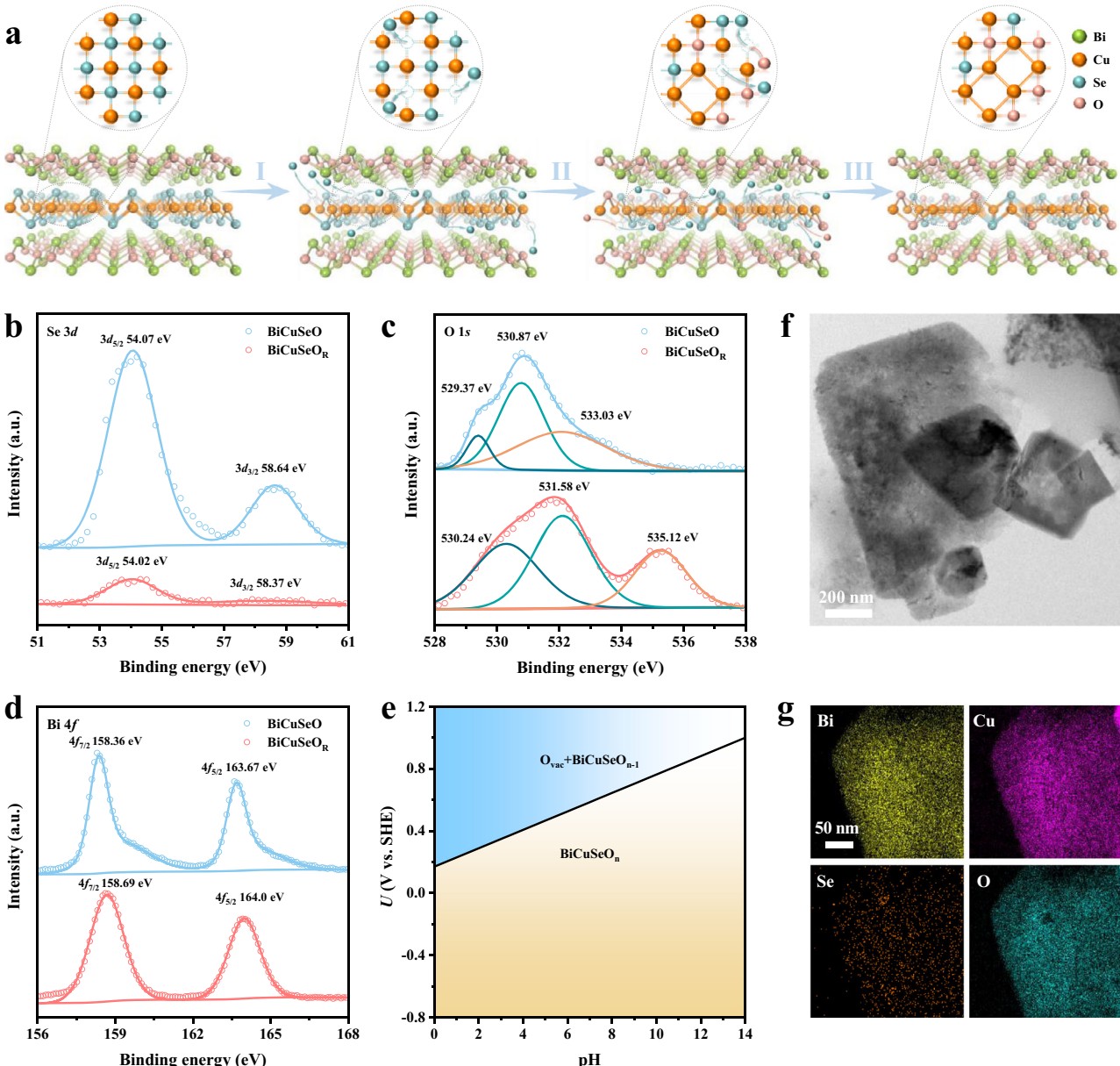

**Fig. 5 Structural evolution and composition analyses of BiCuSeO after CO₂RR. a** Schematic illustration of the in situ structural evolution for BiCuSeO. **b** Se 2*d*, **c** O 1*s*, and **d** Bi 4*f* XPS spectra for BiCuSeO catalysts before and after the CO₂RR. **e** Pourbaix diagram of BiCuSeO. **f** TEM and **g** the corresponding EDS mapping images of BiCuSeO_R.

distribution through the BiCuSeO_R nanosheet in EDS mapping explains the loss of Se (Fig. 5g), which are identical to XAFS and XPS results. Furthermore, the high-resolution O 1*s* XPS spectrum of pristine BiCuSeO (Fig. 5c) can be split into three deconvolution peaks at approximately 529.37, 530.87, and 533.03 eV, which belong to Bi–O, Bi–OH, and surface-adsorbed oxygen species, respectively[26,27]. After CO₂RR, besides the peaks at ~530 eV corresponding to Bi–O and 531.92 eV corresponding to Bi–OH, an intense peak with higher binding energy at 535.29 eV, arising from surface-adsorbed carbonate species[51,52], appears in the O 1*s* spectrum of BiCuSeO_R. These observations are in line with the corresponding Raman results (Fig. 3) and further suggest that the CO₂ molecules are stably adsorbed onto the surface of the BiCuSeO catalyst during the electrocatalytic CO₂RR. As expected, the XPS Bi 4*f* core-level spectra of BiCuSeO_R and BiCuSeO remain consistent (Fig. 5d), which confirms the existence of oxidation state Bi[10,26]. Interestingly, in

addition to changes in the local structure and chemical states, the structural framework of the BiCuSeO superlattice tends to be stable during the CO₂RR, which may be due to the mutual support of the sublayers (Fig. 5). In summary, XANES, EXAFS, XPS, and HRTEM studies consistently confirm that the highly active Bi oxidation state can be stabilized by finely designing superlattices stacked with Bi–O layer and conductive Cu–Se layer, which can undoubtedly contribute to highly selective CO₂ electroreduction performance over a wide potential window and explore its structure-activity relationship. To further understand the stability of $[Bi_2O_2]^{2+}$ sublayer, we calculated a pourbaix diagram to evaluate the potential possibility for surface or subsurface oxygen defects at every pH (0–14) under the reduction conditions (Fig. 5e and Supplementary Fig. S26). According to the calculation, the theoretical formation energy of the surface O defects for the BiCuSeO system was 2.81 eV, suggesting that it's not easy to thermodynamically form surface or subsurface oxygen

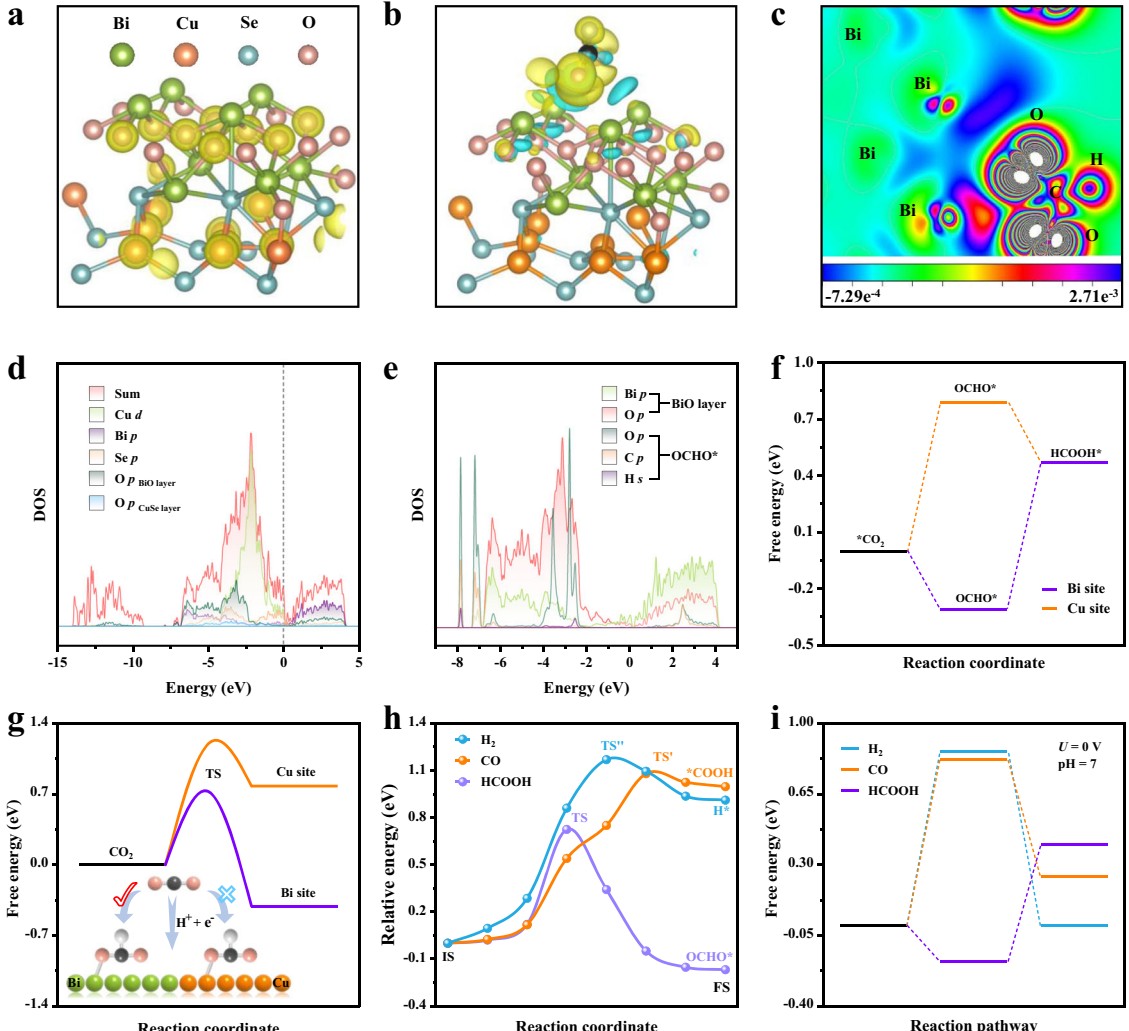

**Fig. 6 DFT calculations. a** Charge density of BiCuSeO after $CO_2RR$. **b** Charge density difference of key reaction intermediates OCHO* on BiCuSeO during the $CO_2RR$. The isosurfaces level is 0.002213 au. The yellow and blue shadows correspond to the charge accumulation and depletion, respectively. **c** Corresponding slice image of Fig. 6b along the (010) face. **d** Total density of states (Sum), Cu *d*, Bi *p*, and Se *p* orbitals PDOS, and O *p* orbitals PDOS obtained, respectively, from BiO and CuSe layers for BiCuSeO during $CO_2RR$ **e** PDOS of *p* orbitals for Bi and O atoms on BiO layer, PDOS of *p* orbitals for O and C atoms from OCHO*, and PDOS of H *s* orbital from the adsorbed OHCO*. **f** The adsorption free energy of OCHO* adsorbed on the Cu and Bi sites. **g** The kinetic energy barrier of adsorbed OHCO*. **h** The relative energy along the formation OCHO*, *COOH and H* species on the BiCuSeO. **i** Gibbs free-energy diagram of $CO_2$ electroreduction to formate, $CO_2$ electroreduction to CO, and HER at pH = 7.

defects. Notably, the pourbaix diagram that oxygen defects would not be generated under the $CO_2RR$ reduction condition.

**DFT calculations**. To explore the intrinsic origin of high formate selectivity and activity over the $[Bi_2O_2]^{2+}$ sublayers in the BiCuSeO superlattice, DFT calculations were performed to investigate the electronic structure of the catalyst as well as the strength of the interaction between the catalytic interface and $CO_2$ molecular species (Supplementary Fig. S27). First, to verify whether the $[Cu_2Se_2]^{2-}$ sublayer of BiCuSeO_R alters and functions as a conductive channel after the structural transformation, their electron charge density, and density of states (DOS) are studied. Clearly, the typical positive charge feature localizes along $[Cu_2Se_2]^{2-}$ sublayers after structural transformation (Fig. 6a), suggesting that $[Cu_2Se_2]^{2-}$ sublayers still maintain a good conductivity and can be employed as conductive channels. The calculation result reveals that the accumulated sublayer charge quantity is −1.44 e (Supplementary Fig. S28), confirming a highly

efficient intersublayer charge separation. Moreover, the total DOS (sum) of BiCuSeO_R in the neighborhood below the Fermi level is mainly contributed by bonding hybridized Cu *d* and Se *p* states from $[Cu_2Se_2]^{2-}$ sublayers (Fig. 6d), which is the origin endowing the conductive character. The sharp peak characteristic in the PDOS of the Cu *d* orbital further illustrates a strong d electron localization, resulting in the localized charge distribution in $[Cu_2Se_2]^{2-}$ sublayers. More notably, different from pristine BiCuSeO reported previously[29], the DOS of BiCuSeO_R below the Fermi level crosses over the Fermi surface, implying a typical semimetallic/metallic nature and thus an enhanced conductivity for BiCuSeO_R after structural evolution. The above results and analyses consistently indicate that $[Cu_2Se_2]^{2-}$ sublayers in the superlattice still maintain fine and even enhanced conductivity. Meanwhile, the PDOS of Bi *p* and Bi *s* orbitals overlap with that of O *p* orbitals to a great extent, indicating a strong interaction between Bi atoms and O atoms, and the oxide state of Bi in the $[Bi_2O_2]^{2+}$ sublayer is negligibly influenced by structural transformation and can be well retained (Supplementary Fig. S29). All

of the above results show that the $[Cu_2Se_2]^{2+}$ sublayer still functions as a conductive channel while $[Bi_2O_2]^{2+}$ sublayer maintains a strong Bi–O coordination structure feature in BiCuSeO superlattices after structural transformation. To further examine the interaction process between catalytic interface and $CO_2$ molecules, we calculated the charge density difference and DOS of OCHO*intermediate adsorbed BiCuSeO$_R$. The charge density difference (Fig. 6b, c) reveals that the charge transfers directly from Bi atoms in $[Bi_2O_2]^{2+}$ sublayers to O atoms in OCHO* intermediate, which is beneficial to activate $CO_2$ molecules, and generate and stabilize formate intermediates OCHO* on the BiCuSeO$_R$ surface. Moreover, OCHO* absorbed BiCuSeO$_R$ displays a higher DOS value near the Fermi level than BiCuSeO$_R$, further suggesting that the good charge transfer from BiCuSeO$_R$ toward formate intermediates OCHO* (Supplementary Fig. S30). Actually, large overlaps of Bi $p$ states and O $p$ states below the Fermi level in the PDOS of OCHO* absorbed BiCuSeO$_R$ (Fig. 6e) further point out that Bi atoms in the $[Bi_2O_2]^{2+}$ layer of BiCuSeO$_R$ have a strong interaction effect with the O atoms in the OCHO* intermediates, which contributes to the absorption and stabilization of the OCHO* intermediate on the BiCuSeO$_R$ surface. Taken together, the specific Bi–O oxide structure of the $[Bi_2O_2]^{2+}$ sublayer endows strong activation capacity of $CO_2$ molecules and stabilization ability of intermediates OCHO*, resulting in high activity for $CO_2$RR.

To better elucidate the high selectivity of the $CO_2$RR, the OCHO* adsorption on the Bi site, Cu site, and Se site are simulated (Supplementary Fig. S31, Table S7, S8)). The calculation results show that OCHO* adsorption on the Se site is weak physical adsorption, so the adsorption of Bi and Cu sites is mainly considered. Clearly, Fig. 6f, g indicates that both the thermodynamic energy barrier and kinetic energy barrier for the formation of OCHO* on Bi site are much lower than that Cu site, which accords with a perfectly linear Brønsted–Evans–Polanyi (BEP) relationship. These results evidently suggest the Bi site is the active site for the $CO_2$RR. Therefore, the reaction Gibbs free energies (ΔG) for OCHO* absorbed on the Bi site based on transient state theory (Supplementary Fig. S32) are calculated and presented in Fig. 6h. For comparison, ΔG for $CO_2$ electroreduction to CO and competitive HER are also performed (Fig. 6h, Supplementary Fig. S33). The calculation results exhibit a much lower energy barrier for the formation of OCHO* (the crucial intermediate) for generating formate compared with CO and HER. These results indicate that BiCuSeO$_R$ kinetically enables the activation of $CO_2$ molecules to form OCHO* intermediate and thus further produce formate (Supplementary Fig. S34). In addition, considering the applied potential and pH effects, the reaction Gibbs free energies under actual work conditions (0.5 M KHCO$_3$, pH = 7) is also calculated (Fig. 6i). Notably, the ΔG value for the OCHO* formation processes is exergonic, indicating their $CO_2$ activation and protonation processes are spontaneous. In contrast, whether $CO_2$ electroreduction to form *COOH or HER to form *H is endergonic, suggests the easy formation of the crucial intermediate OCHO*. Consistent with the predesigned scenario, the DFT calculation results solidly support that in natural BiCuSeO superlattices the $[Cu_2Se_2]^{2-}$ sublayers conduct electrons, while the $[Bi_2O_2]^{2+}$ sublayers act as the active center for the activation of $CO_2$ molecules and subsequent formation/stabilization of OCHO*intermediates, enabling highly selective $CO_2$ electroreduction to formate.

## Discussion

In summary, we propose a tangible active/conductive layer alternately stacked with natural superlattices for stabilizing the metal

oxidation state for high activity and selectivity $CO_2$RR performance. Taking the example of BiCuSeO, which consists of alternately stacked conductive $[Cu_2Se_2]^{2-}$ and insulating $[Bi_2O_2]^{2+}$ sublayers, the comprehensive characterizations reveal that Bi–O layers are retained to drive the activation of $CO_2$ molecules during the $CO_2$RR because the electrons rapidly conducted through conductive $[Cu_2Se_2]^{2-}$ sublayers. Furthermore, DFT calculations indicate that the specific Bi–O coordination in $[Bi_2O_2]^{2+}$ exhibits a strong activating and stabling effect toward the OCHO* intermediate with its Bi p orbitals overlapping with the O p orbitals in OCHO* and enables a highly selective $CO_2$ electroreduction to formate. As a direct outcome, BiCuSeO natural superlattices are found to produce formate with an optimum FE of >90% over a wide potential range from −0.4 to −1.1 V in neutral electrolyte. Our work not only serves as a tangible model with active/conductive layer alternately stacked natural superlattices to suppress the reduction of the metal oxidation electrocatalysts to improve the $CO_2$RR selectivity, but also introduces the specific coordination structures in designing new $CO_2$RR materials.

## Methods

**Synthesis of BiCuSeO Ns**. Bi(NO$_3$)$_2$·5H$_2$O (~95 mg) was first added to a mixed solvent containing 5 mL ultrapure water and 5 mL ethanol followed by ~10 min magnetic stirring. When a milky white suspension was generated, selenourea (26 mg) was added with continued stirring. Then PVP (K30, 100 mg), Cu(NO$_3$)$_2$·3H$_2$O (~50 mg), KOH (120 mg), and NaOH (320 mg) were subsequently added with continuous stirring to obtain a black suspension. Afterward, the contents were transferred into a 50 ml Teflon-lined stainless-steel autoclave and heated at 180 °C for 24 h. After naturally cooling to room temperature, the black precipitate was washed with H$_2$O and ethanol and then naturally dried under ambient conditions.

**Synthesis of Cu$_2$Se Ns**. Cu$_2$Se NS was prepared according to ref. [53]. First, 157 mg Se powder, 5 g NaOH, and 704 mg ascorbic acid were dissolved in 20 mL of H$_2$O at 50 °C, and then added a premade aqueous solution containing 199 mg Cu(CH$_3$-COO)$_2$·H$_2$O, 100 mg beta-cyclodextrin and 30 mL H$_2$O. The mixed solution was placed in a Teflon-lined stainless-steel autoclave and hydrothermally treated at 180 °C for 6 h.

**Synthesis of Bi$_2$O$_3$ Ns**. Bi$_2$O$_3$ Ns was prepared according to the refs. [18,54]. 0.5 mmoL Bi(NO$_3$)$_2$·5H$_2$O was dissolved in 8.5 mL ethylene glycol and 4.3 mL ethanol, and placed in a Teflon-lined stainless-steel autoclave and hydrothermally treated at 160 °C for 5 h.

**Synthesis of Cu$_2$Se/Bi$_2$O$_3$ heterostructure (CuSe-BiO)**. The synthesis process was similar to that of Bi$_2$O$_3$ nanosheeets, but added an amount of Cu$_2$Se nanosheets.

**Characterizations**. The samples were tested by X-ray diffractometer (XRD, Cu Kα, λ = 1.5405 Å, D2 PHASER, Bruker), atomic force microscopy (Dimension Icon, Bruker), X-ray photoelectron spectroscopy (XPS, AXIS-ULTRA DLD-600W), atomic force microscopy (AFM, Dimension Icon, Bruker), confocal Raman system (Alpha 300RS + , WITec), inductively coupled plasma mass spectrometry (ICP-MS, ICPOES730, Agilent), scanning electron microscopy (SEM, FEI quanta 650) and high-resolution transmission electron microscopy (HRTEM, FEI Tecnai G2 F30) equipped with an X-ray energy dispersive spectrometer (EDS). X-ray absorption fine structure spectra (XAFS) were acquired at the BL11B beamline at the Shanghai Synchrotron Radiation Facility (SSRF).

**Electrochemical measurements**. The $CO_2$RR performances of the various samples were tested by using a three-electrode flow cell system in a $CO_2$-saturated 0.5 M KHCO$_3$ aqueous solution. All electrochemical measurements were conducted on a CHI 760E electrochemical workstation (CH instrument, Shanghai, China). And, all the $CO_2$RR performances were measured after the stable LSV scanning. If not specified, all $CO_2$ reduction performance was collected without iR compensation in this work. Approximately 2 mg of catalyst deposited on a gas diffusion layer with a 1 cm$^2$ working area was used as the working electrode (WE). The Ag/AgCl electrode and platinum plate were employed as the reference electrode (RE) and counter electrode (CE), respectively. During the electrochemical measurement, $CO_2$ was pumped into the cathode chamber with a constant flow rate (20 mL min$^{-1}$). The gas products were detected by using an in situ connected gas chromatograph instrument (PANNA, A91lus). The liquid products were

analyzed by nuclear magnetic resonance (NMR) spectrometer (Bruker Ascend TM 600 MHZ). The Faradaic efficiency (FE) during the $CO_2RR$ is calculated by the equation, $FE = Q_i/Q_t = (N_i \times n \times F)/Q_t$, where $Q_i$ is the charge amount for product reduction, $Q_t$ is the total charge consumed, $N_i$ is the product molar amount, n is the electrons transfer number (which is 2 for formate, $H_2$ and CO), and F is the Faradaic constant (96,485 C mol$^{-1}$).

**In situ Raman measurements**. In situ Raman measurements were conducted by employing a top-plate cell euphotic device, which was connected to an electrochemical workstation. The three-electrode (WE, RE, and CE) and electrolyte were the same as those used in the anterior electrochemical measurements. Prior to Raman measurements, $CO_2$ was bubbled into the electrolyte. The in situ Raman spectra were acquired from a confocal Raman spectroscopy (Alpha300, WITec) using a 532 nm laser source.

**In situ XAFS measurements**. In situ XAFS measurements were conducted on homemade equipment (Supplementary Fig. S35). XANES was acquired at the BL11B beamline of the Shanghai Synchrotron Radiation Facility (SSRF), China. The absorption edge position ($E_0$) was calibrated by employing Cu foil, and all the XANES data were collected by fluorescence mode. Every XANES spectrum is tested two or three times.

**DFT calculations**. All the calculations were carried out based on density functional theory (DFT) as implemented in the Vienna Ab initio Software Package code within the Perdew-Burke-Ernzerhof (PBE) generalized gradient approximation and the projected augmented wave (PAW) method[55]. The cutoff energy of the plane-wave basis was 500 eV. The $BiCuSeO_R$ structural models were relaxed by a $3 \times 3 \times 1$ Monkhorst−Pack grid with the solvation correction[56,57]. The convergence criterion for the electronic self-consistent iteration and force was set to $10^{-5}$ eV and 0.01 eV/Å, respectively. A vacuum slap of greater than 30.0 Å was selected to avoid periodic interactions. The Gibbs free-energy change (ΔG) was calculated at 298.15 K according to the computed hydrogen electrode mode[58–60]. The detailed Raman and DFT calculation process and descriptions can be found in Supplementary Note 2, Note 3.

## Data availability

All the data supporting the findings of this study are available within the paper and its Supplementary Information file. The data generated in this study for main manuscript are provided in the Source Data file. All other relevant source data reported in this work are available from the authors on reasonable request. Source data are provided with this paper.

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

## Acknowledgements

This work is financially supported by the National Natural Science Foundation of China (22071069, 21825103, and 22173048), the Hubei Provincial Natural Science Foundation of China (2019CFA002), the Project Funded by China Postdoctoral Science Foundation (2019M662604), Hubei Province Postdoctoral Science and Technology Project, and the Foundation of Basic and Applied Basic Research of Guangdong Province (2019B1515120087). We thank Prof. Gong Penglai (Hebei University) and Dr. Fan Zheng (Shanghai University of Science and Technology) for the helpful discussion of Raman spectroscopy. We also acknowledge technical support from Analytical and Testing Center in Huazhong University of Science and Technology, especially Dr. Zhao Jianquan.

## Author contributions

Y.W.L., J.Y.D. and T.Y.Z. conceived and directed the project. Y.W.L. and J.Y.D. designed the experiments. J.Y.D. carried out the experiments. J.Y.D., T.Y.L., Y.H.Z. and Y.F.L. performed and analyzed the DFT calculations. J.Y.D., R.O.Y., W.B.W. and Y.Z. performed the XAFS and the in situ Raman spectroscopy. Y.W.L. and J.Y.D. co-wrote the paper. H.Q.L. and T.Y.Z. revised the paper. All authors discussed the results and assisted during manuscript preparation. J.Y.D., T.Y.L. and Y.H.Z. contributed equally to this research.

## Competing interests

The authors declare no competing interests.
