## [Peer review file · Nature Communications]

REVIEWER COMMENTS

Reviewer #1 (Remarks to the Author):

Review of 'Active/Conductive Layer Stacked Superlattices for Highly Selective CO₂ Electroreduction'

Junyuan Duan et al. presented a detailed experimental and theoretical study of CO₂ electroreduction to C₁ species (formic acid, syngas) over metal oxide (i.e., BiCuSeO superlattices). They found that [Bi₂O₂]²⁺ remains at the sublayer of BiCuSeO due to the rapid electron transfer between [Bi₂O₂]²⁺ and the conductive [Cu₂Se₂]²⁻ sublayer. Also, from their DFT model, they showed that Bi p orbitals have strong interactions with the O p orbitals in HCOO*, which facilitates the formation of formic instead of syngas. Overall, the paper sounds logically reasonable. There are some main comments below:

1. BiCuSeO is a complex system. The authors should first identify the stability of [Bi₂O₂]²⁺ (i.e., the formation energy of [Bi₂O₂]²⁺) during the reaction, which I didn't see in the current version of the manuscript.
2. Follow up to the previous point, in general, for metal oxide catalysts under the reduction condition, it is very easy to form surface or subsurface oxygen defects, which might most likely be the active site for electrocatalysis. It is very necessary to provide a pourbaix diagram to illustrate the surface defects using phase diagram in the presence of applied potential and pH.
3. Regarding the energy diagram, how the authors calculated the gibbs free energy without even reporting the vibrational frequencies of the reaction intermediates/reactors/products? Authors also didn't report the method of how they calculated the gibbs free energy. Did author use transition state theory? Please be more details. The gibbs free energy seems also didn't take applied potential and pH effects into account there.
4. I didn't see solvation effects in the calculations, which are huge factors to influence the free energy diagram. For formic acid, adsorption/desorption plays an important role during the reduction process, which should consider solvation effects.
5. The energetic calculations are only thermodynamic calculations without kinetic barrier calculations. How authors confirm that the BEP relationship will follow for such complex metal oxide with sublayer active site system?

Reviewer #2 (Remarks to the Author):

see attached.

Reviewer #3 (Remarks to the Author):

Duan et al. reported here a layered structure, featuring BiCuSeO superlattice, as a new electrocatalyst for CO₂RR with 90% formate production within a wide operating potential window. With the aid of in/ex situ spectroscopies and DFT analysis, the authors have suggested the roles of [Bi₂O₂]²⁺ and [Cu₂Se₂]²⁻ sublayers in promoting CO₂-to-formate conversion, though the original proposed structure underwent reconstruction during reaction. I find the catalyst and the mechanism/characterization to be of interest. The paper would require major revision before further consideration.

Major comments:

- The role of Cu-oxide in CO₂RR has been debated and, to my understanding, it has been settled that oxide-derived Cu can present some lasting Cu morphological advantages, BUT the oxide is not present under reducing (operating) conditions. I worry that this paper is re-opening that debate without adding to it substantially.
- The concept is based on the [Cu₂Se₂]²⁻ sublayer transporting electrons and the [Bi₂O₂]²⁺ sublayer serving as active sites. However, as the authors pointed out, most Se atoms (over 90% based on XAS analysis) have leached out during CO₂RR to form a CuSeO compound. How would then the newly derived structure function during CO₂RR compared to the originally proposed one? What is the timescale of this transition? Is the structure formed during CO₂RR still working with the proposed mechanism?
- Bi₂O₃ and Cu₂Se as control samples showed reasonably good performance for formate production (~60%). Does the [Bi₂O₂]²⁺ function as the solely active layer? Or the [Cu₂Se₂]²⁻ derived sublayer and even the [Bi₂O₂]²⁺-[Cu₂Se₂]²⁻ interfaces will also be active sites for formate production?
- The wording in many places is awkward and the paper should be edited by a professional service.

Some minor comments for the authors to consider:

o Page 3, "To investigate the effect of Bi and Cu elements in BiCuSeO for CO₂RR, the electrocatalytic activities of Bi₂O₃ and Cu₂Se Ns (Supplementary Fig. S4 and S5) are also tested for comparisons." Instead, the authors only showed structural characterizations in Fig. S4 and S5.

- o All potentials need to be iR-corrected for better comparison purposes with literature, particularly in Fig. 2f

- o Fig. 2d, all product distributions should be listed in a table in SI, including H₂ and CO.

- o Fig. 2g, Tafel analysis needs to be redone in which tafel slope can be only determined using potentials close to formate onset – a way to avoid the mass transport effect (ACS Catal. 2018, 8, 8121-8129)

- o Fig. 2h, formate FE needs to be shown in addition to H₂ and CO FEs.

- o Page 5, “Noticeably, the overpotential for formate generation is as low as 190 mV, which is smaller than that of most other Bi-based catalysts.” First, the authors should show the calculation of overpotential of formate which is pH-dependent. Second, the authors did not show the evidence of formate onset. Third, the authors should tabulate the claimed performance metrics and compare with literature instead of simply citing other works.

- o The authors should perform ECSA measurements and calculate ECSA-normalized formate current densities to show the increase of intrinsic activities of BiCuSeO.

Duan et al. reported an interesting formate generation using inter-stacking BiCuSeO materials for the CO₂RR. The performance of this reported catalyst seems to outperform the previous Bi-derived catalysts, and such an efficient reactivity is attributed to the stable inter-sublayer charge separation. Bismuthene was reported to be an efficient electrochemical CO₂ reduction catalyst for formic acid generation (Nat Commun 2020, 11, 1088). Bi nanotube could be also synthesized from bismuth oxide for CO₂RR electrocatalysis (Nat Commun 2019, 10, 2807). The intrinsic character of Bi in this proposed BiCuSeO superlattice is still not well characterized and defined. Therefore, the following questions and comments are provided to the authors for revisiting their scientific results and presentation for the future improvement. This manuscript is recommended to undergo a major revision to clarify these uncertainties.

Q1. Can the local Bi-Bi moiety generated under the cathodic condition play the catalytic role for the observed catalytic performance? In Figure 4b, the Bi-Bi signal seems to be observed in the sample of BiCuSeO_R.

Q2. The charge-separated inter-stacking sublayers ([Cu₂Se₂]²⁻ and [Bi₂O₂]²⁺) was attributed to the efficiency of CO₂ electrochemical reduction as stated in line 48-50 of page 2. How this argument is derived is not clear in the cited reference 33-35? The author should rephrase or elaborate this argument.

Q3. If the abovementioned inter-sublayer charge separation plays an important role to CO₂RR, more convincing experimental evidence should be provided to quantitatively measure the accumulated sublayer charge quantity. Additionally, atom relocation is observed during the course of CO₂RR as being reported by the use of BiCuSeO_R sample. Such an atom-relocation phenomenon is intuitively expected to significantly impact the inter-sublayer charge accumulation, subsequently damage the efficiency of CO₂RR. How to rationalize the current density reported in Figure 2h is questionable?

Q4. How BiCuSeO_R model is constructed and rationalized? The underlying scientific rationale of using this BiCuSeO_R model to represent the experimental surface morphology should be elaborated in SI.

Q5. In Figure S19, hydrogen formed at the O sites of BiCuSeO_R model was not characterized computationally. The free energy formation of HER may be overestimated due to the absence of HER on O sites.

Q6. Figure 6g-6i, the plotting of electrochemical step free energy profiles are based on the use of computational electrochemical model. The details of computational methodology for these electrochemical free energy calculations should be elaborated in SI.

Q7. Computed vibrational features for the intermediates using BiCuSeO and BiCuSeO_R models should be integrated (or compared) with the observed in-situ Raman spectrum. That would provide more subtle structural information for the observed spectroscopic signals.

Q8. The formate ions are identified as the dominant product. How do formate ions leave the positive-charged [Bi₂O₂]²⁺ layer, particularly under the minor negative bias potential? Can DFT modeling provide more insights to rationalize this observation?

Comments: The color selection and resolution for plotting Figure 2 (and others) should be improved. Choosing more contradistinctive colors could help the readers to see the subtle difference between these results. High resolution is highly recommended for these pictures.

Response to the Reviewer 1

Overall comments from Reviewer 1's: Review of 'Active/Conductive Layer Stacked Superlattices for Highly Selective CO₂ Electroreduction'

Junyuan Duan et al. presented a detailed experimental and theoretical study of CO₂ electroreduction to C1 species (formic acid, syngas) over metal oxide (i.e., BiCuSeO superlattices). They found that [Bi₂O₂]²⁺ remains at the sublayer of BiCuSeO due to the rapid electron transfer between [Bi₂O₂]²⁺ and the conductive [Cu₂Se₂]²⁻ sublayer. Also, from their DFT model, they showed that Bi p orbitals have strong interactions with the O p orbitals in HCOO*, which facilitates the formation of formic instead of syngas. **Overall, the paper sounds logically reasonable.** There are some main comments below:

Response: We greatly appreciate the approval of our work. In the revised version, new comprehensive experimental data and theoretical calculations were included (These were described in detail below) to address all concerns of the reviewer. The additional data and revisions were marked by yellow highlight background.

Comment 1-1: BiCuSeO is a complex system. The authors should first identify the stability of [Bi₂O₂]²⁺ (i.e., the formation energy of [Bi₂O₂]²⁺) during the reaction, which I didn't see in the current version of the manuscript.

Response: Thanks for the reviewer's insightful comment. Following the reviewer's comment, we calculate the formation energy of [Bi₂O₂]²⁺ (E_{form}) according to the formula $E_{\text{form}} = E_{\text{Bi}_2\text{O}_2} - N_{\text{Bi}}\mu_{\text{Bi}} - N_{\text{O}}\mu_{\text{O}}$. Where $E_{\text{Bi}_2\text{O}_2}$ is the total energy of bulk Bi₂O₂, and N_{Bi} , and N_{O} are the number of Bi and O, respectively. The term μ denotes the chemical potential of species, μ_{Bi} and μ_{O} are taken as atom energy of bulk Bi ($\Delta E_{\text{hull}}=0$) and O₂ molecule. The calculation result shows that the formation energy of [Bi₂O₂]²⁺ in BiCuSeO system is -1.64 eV (**Fig. SR3**), demonstrating that a high stability of [Bi₂O₂]²⁺. Moreover, many experimental studies have shown that the crystalline BiCuSeO with a good stability can be successfully prepared (*Nat. Commun.* 2019, 10, 2814; *Adv. Mater.* 2020, 32, 2003730; *Energy Environ. Sci.* 2017, 10, 1590; *J. Am. Chem. Soc.* 2015, 137, 6587). The above results and analysis also are added in the revised manuscript as **Fig. S1d**.

Fig. SR3 The formation energy of $[\text{Bi}_2\text{O}_2]^{2+}$ during the reaction.

Comment 1-2: Follow up to the previous point, in general, for metal oxide catalysts under the reduction condition, it is very easy to form surface or subsurface oxygen defects, which might most likely be the active site for electrocatalysis. It is very necessary to provide a pourbaix diagram to illustrate the surface defects using phase diagram in the presence of applied potential and pH.

Response: Thanks for the reviewer's insightful comment. We strongly agree with the reviewer's viewpoint that surface oxygen defects might be the active site for electrocatalysis. To further understand the importance of stability and evaluate the role of the oxidation defect, we calculate the potential possibility for surface or subsurface oxygen defects at every pH (0-14) under the reduction condition (**Fig. SR4**). The calculated pourbaix diagram is shown in **Fig. SR5**. According the calculation, the theoretical formation energy of surface O defect for BiCuSeO system is 2.81 eV. Obviously, it is not easy to form to form surface or subsurface oxygen defects thermodynamically. Notably, it can be known from pourbaix diagram that O defects would not be generated under the CO_2RR reduction condition of pH=7 and all negative potentials. The above data and analysis have been updated as **Fig. S26** and **Fig. 5e** in the revised manuscript. We thank the reviewer's kind suggestion again for strengthening the understanding the structural stability.

Fig. SR4 Calculation model of oxygen defects for pourbaix diagram.

Fig. SR5 Pourbaix diagram of BiCuSeO.

Comment 1-3: Regarding the energy diagram, how the authors calculated the gibbs free energy without even reporting the vibrational frequencies of the reaction intermediates/reactors/products? Authors also didn't report the method of how they calculated the gibbs free energy. Did author use transition state theory? Please be more details. The gibbs free energy seems also didn't take applied potential and pH effects into account there.

Response: Thanks for the reviewer's insightful comments. Following the reviewer's comments, we supplemented **Table SR1** containing the vibrational frequencies of the reaction intermediates.

Table SR1 The vibrational frequencies of the reaction intermediates

COOH	$f(\text{cm}^{-1})$	OCHO	$f(\text{cm}^{-1})$	H*	$f(\text{cm}^{-1})$
1f	3479.10	1f	2884.84	1f	1193.50
2f	1542.97	2f	1516.68	2f	571.49
3f	1229.46	3f	1326.90	3f	324.44
4f	1007.19	4f	1310.12		
5f	690.39	5f	1014.67		
6f	628.46	6f	746.23		
7f	287.87	7f	231.92		
8f	226.92	8f	189.60		
9f	200.60	9f	181.63		
10f	140.48	10f	169.24		
11f	99.46	11f	85.49		
12f	80.71	12f	58.39		

Besides, the method for the Gibbs free energy calculation including the using calculation theory, applied potential and pH effects, were provided detailly in Experimental and Supplementary Information (**Supplementary Note 3**). The Gibbs free energy of electrochemical reactions were calculated using the computational hydrogen electrode (CHE), and it could be computed by:

$$\Delta G = \Delta E + \Delta E_{\text{ZPE}} - \Delta TS + \Delta G_{\text{pH}} + \Delta G_{\text{U}} \quad (\text{S1})$$

where E is the ground state energy, E_{ZPE} and S ($T = 298$ K) are the zero-point energy difference and the entropy, respectively. For each system, E_{ZPE} can be calculated by summing vibrational frequencies over all normal modes ν ($E_{\text{zpe}} = 1/2 \sum h\nu$). The effects of pH and electrode potential (U) can be treated as: $\Delta G_{\text{pH}} = 0.0592 \times \text{pH}$ and $\Delta G_{\text{U}} = -eU$. To obtain the proper absolute electrochemical, the work function (Φ) of material should be adjusted by adding electrons. It could be calculated by:

$$U = (W_f - 4.44) + 0.0592 \times \text{pH} \quad (\text{S2})$$

where Φ is the work function relative to reference level, (4.40-0.0592×7) eV is introduced to account for the work function of reversible hydrogen electrode (RHE). According to Neurock methods, the potential-dependent energy can be calculated by:

$$E_{\text{free}}(U) = E_{\text{DFT}} + \int_0^q \langle \bar{V}_{\text{tot}} \rangle dQ + qW_f \quad (\text{S3})$$

Follow the reviewer's comments, we calculate the Gibbs free energy by using transition state theory and the calculation results are shown in **Fig. SR6**. The calculation results exhibit a much lower energy barrier for the formation of OCHO* (the crucial intermediate toward formate) compared with CO and HER. These results indicate that BiCuSeO_R kinetically enables the activation of CO₂ molecules to form OCHO* intermediate and thus further produce formate.

In addition, taking the applied potential and pH effects into account, the reaction Gibbs free energies under actual work condition (0.5 M KHCO₃, pH=7) is also calculated and displayed in **Fig. SR8**. Notably, the Gibbs free energy for OCHO* formation processes is exergonic, indicating their CO₂ activation and protonation processes are spontaneous. By contrast, whether CO₂ electroreduction to form *COOH or HER to form *H are endergonic, further suggesting the favorite formation of the crucial intermediate OCHO*.

Fig. SR6 The Gibbs free energy diagram based on transient state theory.

Fig. SR7 Structural model of intermediates for IS, TS and FS

Fig. SR8 Gibbs free energy diagram of CO₂ electroreduction to formate and CO, and HER at pH=7.

Table SR2. Zero-point energy, enthalpic correction and entropy correction at 298.15 K.

Species	E_{ZPE} (eV)	TS (eV)	G_{corr} (eV)
*COOH	0.6	0.21	0.39
OCHO*	0.6	0.23	0.37
H*	0.13	0.02	0.11

The above data and analysis have been updated as **Fig. 6h, Fig. 6i, Fig. S31, Table S7 and Table S8** in the revised manuscript. We thank the reviewer's kind suggestions again for strengthening the understanding the catalytic mechanism.

Comment 1-4: I didn't see solvation effects in the calculations, which are huge factors to influence the free energy diagram. For formic acid, adsorption/desorption plays an important role during the reduction process, which should consider solvation effects.

Response: Thanks for the reviewer's insightful comment. We totally agreed with the reviewer's points. The solvation effects were considered in the calculations of free energy diagram. To avoid unnecessary misunderstandings, the related instructions about solvation effects have been added in the DFT calculation part the revised manuscript (**Experimental and Supplementary Note 1**).

Comment 1-5: The energetic calculations are only thermodynamic calculations without kinetic barrier calculations. How authors confirm that the BEP relationship will follow for such complex metal oxide with sublayer active site system?

Response: Thanks for the reviewer's insightful comment. We have simulated the OCHO* adsorption on site1 Bi, site 2 Cu, and site 3 Se (**Fig. SR9**). The calculation results showed that OCHO* adsorption on Se site was a weak physical adsorption, so the adsorption on Bi and Cu sites were mainly considered. It was found that Bi site was the active site of CO₂RR, which had a lower reaction energy barrier and satisfied the BEP relationship (**Fig. SR10, SR11**). The above data and analysis have been updated as **Fig. 6f, 6g, Fig. S32** in the revised manuscript.

Fig SR9 Structural model of OCHO* intermediate absorbed on Bi, Cu and Se sites.

Fig. SR10 The thermodynamic energy barrier of adsorbed OCHO*.

Fig. SR11 The kinetic energy barrier of adsorbed OCHO*.

Overall, thank you again for the positive and helpful comments on our theoretical calculations. Under the suggestions, we further sorted out the structure-activity relationship and deepened our understanding of the structural stability of catalysts. Therefore, we updated **Fig. 5** and **Fig. 6** in the revised manuscript.

Fig. 5 Structural evolution and composition analyses of BiCuSeO after CO₂RR. **a** Schematic illustration of the in situ structural evolution for BiCuSeO. **b** Se 2d, **c** O 1s, and **d** Bi 4f XPS spectra for BiCuSeO catalysts before and after CO₂RR. **e** Pourbaix diagram of BiCuSeO. **f** TEM and **g** the corresponding EDS mapping images of BiCuSeO_R.

Fig. 6 DFT calculations. **a** Charge density of BiCuSeO after CO₂RR. **b** Charge density difference of key reaction intermediates OCHO* on BiCuSeO during the CO₂RR. The isosurfaces level is 0.002213 au. The yellow and blue shadows correspond to the charge accumulation and depletion, respectively. **c** Corresponding slice image of Fig. 6b along the (010) face. **d** Total density of states (Sum), Cu d, Bi p and Se p orbitals PDOS, and O p orbitals PDOS obtained, respectively from BiO and CuSe layers for BiCuSeO during CO₂RR **e** PDOS of p orbitals for Bi and O atoms on BiO layer, PDOS of p orbitals for O and C atoms from OCHO*, and PDOS of s orbital from the adsorbed OCHO*. **f** The adsorption free energy of OCHO* adsorbed on the Cu and Bi sites. **g** The kinetic energy barrier of adsorbed OCHO*. **h** The relative energy along the formation OCHO*, *COOH and H* species on the BiCuSeO. **i** Gibbs free energy diagram of CO₂ electroreduction to formate, CO₂ electroreduction to CO, and HER at pH=7.

Response to the Reviewer 2

Overall comments from Reviewer 2's: Duan et al. reported an interesting formate generation using inter-stacking BiCuSeO materials for the CO₂RR. The performance of this reported catalyst seems to outperform the previous Bi-derived catalysts, and such an efficient reactivity is attributed to the stable inter-sublayer charge separation. Bismuthene was reported to be an efficient electrochemical CO₂ reduction catalyst for formic acid generation (Nat Commun 2020, 11, 1088). Bi nanotube could be also synthesized from bismuth oxide for CO₂RR electrocatalysis (Nat Commun 2019, 10, 2807). The intrinsic character of Bi in this proposed BiCuSeO superlattice is still not well characterized and defined. Therefore, the following questions and comments are provided to the authors for revisiting their scientific results and presentation for the future improvement. This manuscript is recommended to undergo a major revision for clarify these uncertainties.

Response: We greatly appreciate the approval of our work. For the literature recommended by reviewers, zero-valence metal Bi exhibit excellent activity CO₂ reduction, but their CO₂RR activity including other reports in Fig. 2f are difficult to maintain in a wide potential window due to the competitive hydrogen evolution reaction (HER) performance. Metal oxides are tightly correlated with the high selectivity towards CO₂RR. Due to the unavoidable self-reduction to zero-valence metal, developing electrocatalysts based on metallic oxidation state rather than their self-reduction is highly desirable for performance optimization. In response, our manuscript proposes a tangible structural model of natural superlattices, i.e. BiCuSeO, in which the conductive [Cu₂Se₂]²⁻ sublayer rapidly conducts electrons to protect the active center of the [Bi₂O₂]²⁺ sublayer to drive the activation of CO₂ molecules. Gratifyingly, BiCuSeO nanosheets exhibit a high catalytic selectivity featured by formate Faradaic efficiency of > 90% at a wide potential range from -0.4 to -1.1 V. Therefore, the focus of this manuscript is to construct metallic oxidation state (Bi-O) rather than zero-valence metal (Bi) toward high CO₂RR activity. In addition, new comprehensive experimental data are included (describe in detail below) to address all concerns of the reviewer in the revised version. The additional data and revisions were marked by yellow highlight background.

Comment 2-1: Can the local Bi-Bi moiety generated under the cathodic condition play the

catalytic role for the observed catalytic performance? In Figure 4b, the Bi-Bi signal seems to be observed in the sample of BiCuSeO_R.

Response: Thanks for the reviewer's insightful comments. As you concerned, Bi-Bi signal is indeed observed in the XANES spectra (**Fig. 4a**). Compared with the pristine BiCuSeO, the absorption edge of BiCuSeO_R in XANS spectra slightly shifts towards a lower energy (**Fig. 4a**). Following the reviewer's critical comments, we conducted linear combination fitting (LCF) of XANES spectrum of BiCuSeO_R, shown in **Fig. SR12**. It can be found that the content of metal Bi is less than 5% (4.4%), indicating that the oxidation state of Bi mainly retains. In fact, Bi-Bi signals in the metallic Bi powder exhibit characteristic double-peaks located at ~2.58 and ~3.16 Å in R space (**Fig. 4b**). The single peak appearing at ~3.01 Å for BiCuSeO_R can mainly be attributed to the first coordination shell of Bi-Se bond. Correspondingly, Bi-Bi coordination number is only 0.3 according to EXAFS fitting (**Table S8**). Moreover, the WT contour plots of BiCuSeO_R display only one intensity maximum at ~1.6 Å, which is apparently corresponding to the Bi-O coordination with BiCuSeO_R rather than the Bi-Bi in metallic Bi (**Fig. 4e**). All of the above results shows that only a very little Bi-Bi is present and Bi-O structures are mainly retained.

Numerous researches indicated that metallic electrodes would promote the competing hydrogen evolution reaction (HER), and hence lowered the CO₂RR selectivity. In contrast, metal oxides were proved to be beneficial for promoting the reaction kinetics and selectivity towards CO₂RR. BiCuSeO_R superlattices exhibit a high catalytic selectivity featured by formate Faradaic efficiency FE of >90% at a wide potential range from -0.4 to -1.1 V, while the FE for CO and H₂ gas are ~2.4% and ~2.4%, respectively (**Fig. 2c**). FE_{formate} of for BiCuSeO are much higher than that of Cu₂Se (the maximum FE_{formate} is ~60%) and Bi₂O₃ (the maximum FE_{formate} is ~85%) at the tested potentials, indicating that BiCuSeO_R Ns is more inclined to yield the formate product (**Fig. 2d**). It is reasonably believed that the Bi-O moiety plays a key role in catalytic performance rather than Bi-Bi moiety.

Fig. SR12 XANES spectrum and linear combination fitting results of BiCuSeO_R

Comment 2-2: The charge-separated inter-stacking sublayers ($[\text{Cu}_2\text{Se}_2]^{2-}$ and $[\text{Bi}_2\text{O}_2]^{2+}$) was attributed to the efficiency of CO₂ electrochemical reduction as stated in line 48-50 of page 2. How this argument is derived is not clear in the cited reference 33-35? The author should rephrase or elaborate this argument.

Response: We greatly appreciate the reviewer's insightful concern and feel deeply sorry for the inconvenience caused to the reviewers by our incomplete description in manuscript. The cited reference 33-35 in the original manuscript point to that natural BiCuSeO superlattice consists of conductive $[\text{Cu}_2\text{Se}_2]^{2-}$ and insulating $[\text{Bi}_2\text{O}_2]^{2+}$ sublayers stacking alternately along *c*-axis, which fits our design concept. Therefore, to avoid unnecessary misunderstandings, we move the reference to the statement "conductive $[\text{Cu}_2\text{Se}_2]^{2-}$ and insulating $[\text{Bi}_2\text{O}_2]^{2+}$ sublayers stacking alternately along *c*-axis" as Ref. 29-31 in the revised manuscript.

Comment 2-3: If the abovementioned inter-sublayer charge separation plays an important role to CO₂RR, more convincing experimental evidence should be provided to quantitatively measure the accumulated sublayer charge quantity. Additionally, atom relocation is observed during the course of CO₂RR as being reported by the use of BiCuSeO_R sample. Such an atom-relocation phenomenon

is intuitively expected to significantly impact the inter-sublayer charge accumulation, subsequently damage the efficiency of CO₂RR. How to rationalize the current density reported in Figure 2h is questionable?

Response: Thanks for the reviewer's comment. As the reviewer was concerned, atom relocation is observed during the course of CO₂RR as being reported by the use of BiCuSeO_R sample. Therefore, the structural model of our theoretical simulation is BiCuSeO_R. The results explain that Cu-Se sublayers and Bi-O sublayers in the BiCuSeO_R can still play the functions of conductive layer and active layer, respectively. Moreover, in the experimental, all the CO₂ RR performances in the manuscript were measured after the stable LSV scanning, the transformation from BiCuSeO to BiCuSeO_R was considered to be accomplished. Therefore, the current density reported in Figure 2h attribute to BiCuSeO_R, which is consistent with the result of our theoretical calculation.

Fig. SR13 Inter-sublayer charge separation.

Following the comment, the inter-sublayer charge separation of BiCuSeO_R was also calculated and the results were shown in **Fig. SR13**. The calculation result reveals that [Cu₂Se₂]²⁻ sublayer can efficiently capture electrons, while the [Bi₂O₂]²⁺ sublayer are depleted of electrons. And the accumulated sublayer charge quantity is -1.44 e. All the above results suggest that the conductive

$[\text{Cu}_2\text{Se}_2]^{2-}$ sublayer efficiently conducts electrons to protect the active center of the $[\text{Bi}_2\text{O}_2]^{2+}$ sublayer to drive the activation of CO_2 molecules during the electrochemical CO_2RR . The above data and analysis have been updated as **Fig. S28** in the revised manuscript.

Comment 2-4: How BiCuSeO_R model is constructed and rationalized? The underlying scientific rationale of using this BiCuSeO_R model to represent the experimental surface morphology should be elaborated in SI.

Response: Thanks for the reviewer's good comment. From the XAS, TEM, and XPS analysis results, BiCuSeO_R also maintain the crystalline phase, morphologies and structure of tetragonal BiCuSeO . The construction process of BiCuSeO_R is simply proposed in **Fig. 5a**. During the electrocatalytic CO_2RR process, the conductive Cu_2Se_2 layer are firstly interplayed by the negative potential and the Se atoms instantly escaped from the Cu_2Se_2 sublayers. With breaking of Cu-Se bonds, partial Se atoms are replaced in situ by the strong electronegative O to form Cu-O bonds, at the same time Cu-Cu bonds are generated locally. Due to the confinement of Bi_2O_2 layer, the frame of Cu_2Se_2 layer tend to be stable. Meanwhile, the Bi_2O_2 sublayer are protected and mainly kept because the conductive Cu_2Se_2 sublayer rapidly conducts electrons. The underlying scientific rationale of using this BiCuSeO_R model to represent the experimental surface morphology has been elaborated in SI (This is updated as **Supplementary Note 1**).

Fig. 5a Schematic illustration of the in situ structural evolution for BiCuSeO .

Comment 2-5: In Figure S19, hydrogen formed at the O sites of BiCuSeO_R model was not characterized computationally. The free energy formation of HER may be overestimated due to the absence of HER on O sites.

Response: Thanks for the reviewer's comment. According to structural model optimization by theoretical calculation, hydrogen cannot be adsorbed at the O site of BiCuSeO_R model. Hydrogen tends to be adsorbed at the Bi site, and the free energy formation of HER calculated by using the corresponding structural model (**Fig. SR14**). The above data and analysis have been updated as **Fig. S33** in the revised manuscript.

Fig. SR14 Calculated hydrogen adsorbed on BiCuSeO_R model.

Comment 2-6: Figure 6g-6i, the plotting of electrochemical step free energy profiles are based on the use of computational electrochemical model. The details of computational methodology for these electrochemical free energy calculations should be elaborated in SI.

Response: Thanks for the reviewer's comment. Following the useful suggestions, the details of computational methodology for these electrochemical free energy calculations have been elaborated in SI (**Supplementary Note 3**). For example, energy optimization method (Perdew-Burke-Ernzerhof (PBE) generalized gradient approximation (GGA) and the projected augmented wave (PAW)), the cutoff energy (500 eV), and the Brillouin zone for geometry optimization and electronic structure computations ($3 \times 3 \times 1$ and $4 \times 4 \times 1$), the convergence criterion for the electronic self-consistent iteration and force (10^{-5} eV and 0.01 eV/Å), the distance of vacuum space (30.0 Å), the van der Waals (vdW) interactions method, Poisson-Boltzmann implicit solvation model, the reaction energy barrier simulation method (climbing-image nudged elastic band (CI-NEB) method), the formulas and methods for the Gibbs free energy calculation. The detail descriptions are supplemented in **Supplementary Note 3**.

Comment 2-7: Computed vibrational features for the intermediated using BiCuSeO and BiCuSeO_R models should be integrated (or compare) with the observed in-site Raman spectrum. That would provide more subtle structural information for the observed spectroscopic signals.

Response: Thanks for the reviewer's insightful comment. Following the comment, we calculated the Raman vibrational features for the intermediates *CO₂ and OCHO*. The calculated Raman peaks of asymmetric C–O stretching vibration modes for OCHO* is located at 1544.98 cm⁻¹ (Table SR3 and Fig. SR15b), which agreed well the experimental Raman measurement (1540 cm⁻¹). The calculated Raman vibration peak of *CO₂ was located at 1295.1 cm⁻¹, which can be correspond to the experimental test of *CO₂ at 1160 cm⁻¹ (Table SR4 and Fig. SR15a). The difference between theoretical and experimental values might be caused by the electrolyte environment and complex structure vibration of CO₂ molecules. The above data and analysis have been updated as Fig. S12, Table S5 and Table S6 in the revised manuscript.

Fig. SR15 Computed Raman vibrational features for the intermediates using BiCuSeO model

Table SR3 Computed vibrational features for the intermediate OCHO*

Raman shift (cm ⁻¹)	990.23	1252.92	1324.14	1544.98	2458.98
Vibrational feature					

Table SR4 Computed vibrational features for the intermediate *CO₂

Raman shift (cm ⁻¹)	593.81	593.81	1295.1	2458.98
Vibrational feature				

Comment 2-8: The formate ions are identified as the dominate product. How does formate ions leave the positive-charged [Bi₂O₂]²⁺ layer, particularly under the minor negative bias potential? Can DFT modeling provide more insights to rationalize this observation?

Response: Thanks for the reviewer's comment. We used DFT calculations to identify the formate leaving from the positive-charged [Bi₂O₂]²⁺ layer (**Fig. SR16**). It can be seen that formate is produced by reducing OCHO* with one electron and proton transfer according to the formula OCHO* + H⁺ + e⁻ = HCOOH. From the DFT calculation, HCOOH is electrically neutral and exhibited a physical adsorption of formate on the positive-charged [Bi₂O₂]²⁺ layer. Moreover, the Gibbs free energy of the desorption for formate from [Bi₂O₂]²⁺ layer is -0.07 eV, indicating formate desorption processes are spontaneous. Taken together, it was very easy for the desorption of formate without extra energy. The above data and analysis have been updated as **Fig. S34** in the revised manuscript.

Fig. SR16 DFT calculation of formate producing and leaving.

Comment 2-9: The color selection and resolution for plotting Figure 2 (and others) should be improved. Choosing more contradistinctive colors could help the readers to see the subtle difference between these results. High resolution is highly recommended for these pictures.

Response: Thanks for the reviewer's comment. The more contradistinctive colors selection and high resolution for plotting Fig. 2 were used. Besides, the resolution for all figures were improved.

Response to Reviewer #3

Overall comment from Reviewer 3's: Duan et al. reported here a layered structure, featuring BiCuSeO superlattice, as a new electrocatalyst for CO₂RR with 90% formate production within a wide operating potential window. With the aid of in/ex situ spectroscopies and DFT analysis, the authors have suggested the roles of [Bi₂O₂]²⁺ and [Cu₂Se₂]²⁻ sublayers in promoting CO²⁻ to-formate conversion, though the original proposed structure underwent reconstruction during reaction. **I find the catalyst and the mechanism/characterization to be of interest.** The paper would require major revision before further consideration.

Response: We greatly appreciate the approval of our work. In the revised version, new comprehensive experimental data are included (are described in detail below) to address all concerns of the reviewer. The additional data and revisions were marked by yellow highlight background.

Comment 3-1: The role of Cu-oxide in CO₂RR has been debated and, to my understanding, it has been settled that oxide-derived Cu can present some lasting Cu morphological advantages, BUT the oxide is not present under reducing (operating) conditions. I worry that this paper is re-opening that debate without adding to it substantially.

Response: Thanks for the reviewer's insightful comments. As stated by the reviewer, copper oxide derived Cu⁰/Cu⁺ species played the crucial role of the production of C₂₊ products. In this manuscript, we concerned with metal oxides (such as SnO₂, Bi₂O₃, In₂O₃) with formate production activity. In addition, as we emphasized in the introduction, we concluded that the oxidation state of metal was the active sites for the CO₂RR into formic acid. However, metal oxides would undergo in situ self-reduction to metallic states during the CO₂RR, and thus enhancing the competitive hydrogen evolution reaction (HER). The focus of our work was developing electrocatalysts based on metallic oxidation state to obtain highly efficient formic acid. Along this avenue, we proposed a tangible structural model of natural superlattices, i.e. BiCuSeO, by using the conductive [Cu₂Se₂]²⁻ sublayer to rapidly conducts electrons and thus protect the active center of the [Bi₂O₂]²⁺ sublayer, driving the activation of CO₂ molecules.

Comment 3-2: The concept is based on the $[\text{Cu}_2\text{Se}_2]^{2-}$ sublayer transporting electrons and the $[\text{Bi}_2\text{O}_2]^{2+}$ sublayer serving as active sites. However, as the authors pointed out, most Se atoms (over 90% based on XAS analysis) have leached out during CO_2RR to form a CuSeO compound. How would then the newly derived structure function during CO_2RR compared to the originally proposed one? What is the timescale of this transition? Is the structure formed during CO_2RR still working with the proposed mechanism?

Response: Thanks for the reviewer's critical comments. Although most Se atoms have escaped during CO_2RR process, partial O atoms entered synchronously, Bi-O sublayer has been well maintained (**Fig. 4**), the crystalline BiCuSeO phase can be mainly retained and the structural framework of BiCuSeO superlattice tends to be stable (**Fig. 4 and Fig. 5**). Interestingly, the newly derived structure of BiCuSeO nanosheets after the CO_2RR (BiCuSeO_R) topologically inherited the primitive BiCuSeO superlattice structure. Moreover, the function for the CO_2RR were clarified by detailed theoretical calculation and the structural model is the BiCuSeO_R in our manuscript. In detail, the total DOS of BiCuSeO_R in the neighborhood below the Fermi level is mainly contributed by bonding hybridized Cu d and Se p states from $[\text{Cu}_2\text{Se}_2]^{2-}$ sublayers and crossing over Fermi surface, implying that $[\text{Cu}_2\text{Se}_2]^{2+}$ sublayer still maintain fine and even enhanced conductivity. Meanwhile, PDOS of BiCuSeO_R also indicated a strong interaction between Bi atoms and O atoms in $[\text{Bi}_2\text{O}_2]^{2+}$ sublayer. The structural characterizations and DFT calculations consistently showed that that $[\text{Cu}_2\text{Se}_2]^{2+}$ sublayer still functions as a conductive channel while $[\text{Bi}_2\text{O}_2]^{2+}$ sublayer maintain strong Bi-O coordination structure feature in BiCuSeO_R after structural transformation. Doubtlessly, BiCuSeO_R play the same function as the primary BiCuSeO originally proposed in **Fig. 1a** during CO_2RR . Therefore, in BiCuSeO_R , the conductive $[\text{Cu}_2\text{Se}_2]^{2-}$ sublayer rapidly conducts electrons to protect the active center of the $[\text{Bi}_2\text{O}_2]^{2+}$ sublayer to drive the activation of CO_2 molecules.

In the experimental, we firstly scanned linear sweep voltammetry (LSV) curves. When the LSV curves was stable without additional redox peaks, the transformation from BiCuSeO to BiCuSeO_R was considered to be accomplished. The whole process needed roughly 6-10 times LSV cycles (~20-30 min). And, all the CO_2RR performances in the manuscript were measured after the stable LSV scanning. These experimental details have been added in the revised manuscript.

Comment 3-3: Bi₂O₃ and Cu₂Se as control samples showed reasonably good performance for formate production (~60%). Does the [Bi₂O₂]²⁺ function as the solely active layer? Or the [Cu₂Se₂]²⁻ derived sublayer and even the [Bi₂O₂]²⁺-[Cu₂Se₂]²⁻ interfaces will also be active sites for formate production?

Response: Thanks for the reviewer's important comments. Cu₂Se showed passable performance for formate production (~60%), which is consistent with literature reports. Meanwhile, we further calculate the free energy at the Cu site and possessed considerable energy barrier relative to Bi site on the [Bi₂O₂]²⁺. Therefore, theoretical simulation and experiment verification are combined to conclude the inadequate activity of [Cu₂Se₂]²⁻. Bi₂O₃ show reasonably good performance for formate production (the maximum FE_{formate} is ~85%), which is agree with literature reports. However, it's worth noting that Bi₂O₃ undergo in situ self-reduction to zero-valence Bi during the CO₂RR. Therefore, with this self-reduction, the competitive hydrogen evolution reaction (HER) performance of the derived metal catalysts will gradually dominate¹, resulting in their CO₂RR activity are difficult to maintain over a wide potential window. The performance and structural tests in our manuscript confirmed the above views. To resolve this contradictory of spontaneous self-reduction of metal oxides and the high selective CO₂RR performance, we propose a tangible structural model of natural superlattices, i.e. BiCuSeO, in which the conductive [Cu₂Se₂]²⁻ sublayer rapidly conducts electrons to protect the active center of the [Bi₂O₂]²⁺ sublayer to drive the activation of CO₂ molecules. In situ characterization and theoretical calculation also confirms this hypothesis, that is, Bi-O coordination in [Bi₂O₂]²⁺ exhibits a strong coupling effect with its Bi p orbitals overlapping with O p orbitals in OCHO* and enables a highly selective CO₂RR to formate. Meanwhile, the [Cu₂Se₂]²⁻ main contributes as electronic channel. Benefitting by the interplay of Cu-Se layer/Bi-O layer, natural BiCuSeO superlattices exhibit a high catalytic selectivity featured by formate Faradaic efficiency FE of >90% at a wide potential range from -0.4 to -1.1 V.

Comment 3-4: The wording in many places is awkward and the paper should be edited by a professional service.

Response: Thanks for the reviewer's useful suggestion. To resolve our language issues and meet the standards of *Nat. Commun.*, the entire manuscript was edited by a professional science editor

from ACS Authoring Services. The certificate from the editing service was also attached in resubmitted manuscript (**Fig. SR17**). This issue may have led to the misunderstanding of the reviewer and editor. Therefore, we apologize for this careless mistake again. The changes of grammar correction and language polishing in the revised manuscript were marked by red color.

Fig. SR17 The certificate for language editing.

Comment 3-5: Page 3, “To investigate the effect of Bi and Cu elements in BiCuSeO for CO₂RR, the electrocatalytic activities of Bi₂O₃ and Cu₂Se Ns (Supplementary Fig. S4 and S5) are also tested for comparisons.” Instead, the authors only showed structural characterizations in Fig. S4 and S5.

Response: Thanks for the reviewer’s earnest comment. Our expression in the original manuscript is insufficiently rigorous. To avoid misunderstanding, we have revised our statements in the revised manuscript and the corresponding corrections are identified in yellow highlight background in the revised manuscript. “To investigate the effect of Bi-O and Cu-Se sublayers in BiCuSeO for the CO₂RR, Bi₂O₃, Cu₂Se Ns and Cu₂Se-Bi₂O₃ heterostructures (CuSe-BiO) are prepared, and their structural characterizations are displayed in **Supplementary Fig. S3, S4 and S5**. The corresponding electrocatalytic activities are also tested for comparisons, as shown in **Fig. 2**”

Comment 3-6: All potentials need to be iR-corrected for better comparison purposes with literature, particularly in Fig. 2f

Response: Thanks for the reviewer's comment. The CO₂RR performance with iR compensation (85%) were tested and shown in **Fig. SR18**. The results showed that a maximum current density of ~ 219 mA cm⁻² was obtained at -1.0 V (Fig. SR10a). Moreover, BiCuSeO also maintained an outstanding formate selectivity over a wide potential window with iR compensation (**Fig. SR17b**). Considering that the solution resistance cannot be eliminated in the actual electrolysis process (such as *Energy Environ. Sci.* **2018**, *11*, 744-771.; *Chem. Soc. Rev.* **2020**, *49*, 9154-9196.), we still present CO₂RR performance without iR compensation in the Fig. 2. Meanwhile, the iR-corrected CO₂RR performance was also added in the Supplementary Information as **Fig. S7**.

Fig. SR18 CO₂RR performance of BiCuSeO in 0.5 M KHCO₃ a LSV curves with and without iR compensation. b FE_{formate} with iR compensation.

Comment 3-7: Fig. 2d, all product distributions should be listed in a table in SI, including H₂ and CO.

Response: Thanks for the reviewer's good suggestions. Following the reviewer's comment, we supplemented the **Table SR5** that listed all product distributions including formate, CO and H₂. This table has been updated as **Table S1** in the Supplementary Information.

Table SR5 CO₂ RR product distributions for BiCuSeO, Cu₂Se and Bi₂O₃

E (V)	FE_{formate}(%)				FE_{CO}(%)				FE_{H₂}(%)			
	BiCuSeO	Cu ₂ Se	Bi ₂ O ₃	CuSe-BiO	BiCuSeO	Cu ₂ Se	Bi ₂ O ₃	CuSe-BiO	BiCuSeO	Cu ₂ Se	Bi ₂ O ₃	CuSe-BiO
-0.4	93.74	46.5	81.5	52.1	0	0.32	0	0.3	1.45	18.11	5.51	47.5
-0.5	93.42	57	80.7	72.3	0.02	0.28	0.06	0.3	6.56	33.91	12.81	18.4
-0.6	90.25	52.1	83.5	62.4	0.11	0.14	0.23	0.6	9.64	37.13	11.77	35.5
-0.7	90.29	50.7	84.6	69.1	6.21	0.43	0.62	2.2	3.5	48.07	16.3	27.8
-0.8	92.9	61.05	86.2	81.4	2.4	0.87	2.55	2	4.7	28.98	13.05	7.3
-0.9	93.4	54.92	84.8	89.8	2.36	0.51	3.21	2.9	3.35	41.42	12.91	8.3
-1.0	90.02	53.27	82.3	71.4	3.87	2.52	2.26	2.9	6.11	43.61	13.02	15.5
-1.1	90.24	44.08	81.4	52.6	2.75	6.24	3.33	2.1	7.01	44.57	15.55	28.7

Comment 3-8: Fig. 2g, Tafel analysis needs to be redone in which tafel slope can be only determined using potentials close to formate onset-a way to avoid the mass transport effect (ACS Catal. 2018, 8, 8121-8129)

Response: Thanks for the reviewer's useful comment. Following the reviewer's comment, Tafel slope has been redone using potentials close to formate onset-a way to avoid the mass transport effect (ACS Catal. 2018, 8, 8121-8129). As shown in **Fig. SR19**, the revised Tafel slope of BiCuSeO were fit with the theoretical model of electron transfer during CO₂RR into formate and still smaller than the contrast samples. We acknowledge again for the valuable reminder, the kinetic studies of CO₂RR are clearer and more solid. And the revised Tafel slope have been updated as **Fig. 2g** in the revised manuscript.

Fig. SR19 Tafel slopes.

Comment 3-9: Fig. 2h, formate FE needs to be shown in addition to H₂ and CO FEs.

Response: Thanks for the reviewer's useful suggestions. Following the reviewer's comment, we supplemented formate FE and the FE_{formate} are all over 90% in different time periods in Fig. SR20. The revised chronoamperometry have been updated as Fig. 2h in the revised manuscript.

Fig. SR20 10 h Chronoamperometry results for BiCuSeO at -0.9 V.

Comment 3-10: Page 5, “Noticeably, the overpotential for formate generation is as low as 190 mV, which is smaller than that of most other Bi-based catalysts.” First, the authors should show the calculation of overpotential of formate which is pH-dependent. Second, the authors did not show the evidence of formate onset. Third, the authors should tabulate the claimed performance metrics and compare with literature instead of simply citing other works.

Response: Thanks for the reviewer’s important comment. Following the reviewer’s suggestions, we tested the CO₂ RR performance of BiCuSeO nanosheets in 0.5 M KHCO₃ solution (1.0 atm, 25 °C, and pH = 7.0) under the potentials of -0.3 V, -0.35 V and -0.4 V (RHE) again. The liquid products were collected and tested by NMR and shown in **Fig. SR21**. NMR spectra, clearly showing that formate can be produced at -0.4 V. Moreover, weak signal of formate seemed to be occurred at the NMR spectrum of product obtained at -0.35 V. This result displayed that formate can be generated at a relatively potential of -0.4 V.

According to the report published in *Adv. Mater.*, **32**, 2001848 (2020), the thermodynamic potential for formate production from CO₂RR reaction in aqueous electrolyte under standard conditions (1.0 atm, 25 °C, and pH = 7.0) is about -0.21 V. So, the overpotential for formate generation can be calculated to be about 190 mV.

To highlight performance metrics and show clear comparisons with literature, the formate performance of various samples was summarized in the **Table SR6**. The above data and analysis have been updated as **Fig. S6** and **Supplementary Table S2** in the revised manuscript.

Fig. SR21 ¹H NMR spectra of the liquid product at different potential

Table SR6. CO₂RR performances comparison of Bi-based electrocatalysts for formate production

Samples	FE _{max} /E(V)	FE _{min} /E(V)	E _{max} (V)	E _{min} (V)	FE _[E_{min}]	Overpotential (mV)	iR (Y/N/U)	References
1 β -Bi ₂ O ₃ fractals	87%/-1.2	27%/-0.8	-1.2	-0.8	27%	590	N	Adv. Funct. Mater. 2020, 30, 1906478
2 Bi Ns	86%/-1.1	33%/-0.5	-1.2	-0.5	33%	290	N	Nano Energy 2018, 53, 808-816
3 Bi ₂ O ₃ sphere	90%/-0.9	62%/-0.7	-1.1	-0.7	62%	490	U	ACS Catal. 2020, 10, 1, 743-750
4 Bi ₂ O ₃ NPs/C	93%/-0.9	80%/-0.7	-1.1	-0.7	80%	490	U	Angew. Chem. Int. Ed. 2020, 59, 10807-10813
5 BiPO ₄ Ns	92%/-0.9	74%/-0.8	-1.2	-0.8	74%	590	U	Angew. Chem. Int. Ed. 2021, 60, 7681-7685
6 Bi ₂ O ₃ Ns/CNT	93.6%/-1.2 56	14%/-0.556	-1.356	-0.556	14%	346	N	Angew. Chem. Int. Ed. 2019, 58, 13828-13833
7 Bi Ns	100%/-0.7	3%/-1.1	-1.1	-0.7	100%	490	Y	Angew. Chem. Int. Ed. 2020, 59, 20112-20119
8 Bi NTs	97%/-0.85	5%/-0.38	-1.05	-0.38	5%	170	N	Nat. Commun. 2019, 10, 2807
9 Bi Ns	95%/-0.9	18%/-0.603	-1.06	-0.51	<1%	300	N	Nat. Commun. 2018, 9, 1320
10 Bi dendrite	~89%/-0.7 4	29%/-1.18	-1.18	-0.56	64%	350	U	ACS Catal. 2017, 7, 8, 5071-5077
11 Bi NWs	95%/-0.7	41%/-0.5	-1.19	-0.5	41%	290	Y	Energy Environ. Sci., 2019, 12, 1334-1340
12 Bi nanoflakes	100%/-0.9	79.5%/-0.4	-1.2	-0.4	79.50%	190	N	Nano Energy 2017, 39, 44-52
13 BiO _x /C	97%/-0.92	22%/-0.65	-1.4	-0.65	22%	440	Y	ACS Catal. 2018, 8, 2, 931-937
14 Bi Ns	99%/-0.9	56%/0.75	-0.95	-0.75	56%	540	Y	Adv. Mater. 2018, 30, 1802858
15 Bi-Sn/CF	96%/-1.14	44%/-0.64	-1.24	-0.64	44%	420	U	Adv. Energy Mater. 2018, 8, 1802427
16 BiCuSeO	93.4%/-0.9	90.02%/-1.0	-1.1	-0.4	~93%	190	N	This work ★

Note: Y: with iR compensation; N: without iR compensation; U: undefined.

Comment 3-11: The authors should perform ECSA measurements and calculate ECSA-normalized formate current densities to show the increase of intrinsic activities of BiCuSeO.

Response: Thanks for the reviewer's useful suggestions. Following the reviewer's comments, we tested the cyclic voltammetry (CV) curves with different scan rates in a non-Faradic region of -0.295~ 0.345 V versus Ag/AgCl (**Fig. SR22**). The corresponding electrochemical double electric layer capacitances (C_{dl}) of BiCuSeO, Cu_2Se and Bi_2O_3 were obtained by the equation $C_{dl} = \Delta j/v$ (**Fig. SR23**), where Δj and v are current density difference and scan rates respectively. Accordingly, the ESCA values of BiCuSeO, Cu_2Se and Bi_2O_3 were calculated by the formula $ESCA = C_{dl}/C_s$ ($C_s=60 \mu F cm^{-2}$), which were showing in **Table SR7**. Correspondingly, the ECSA-normalized formate current densities of BiCuSeO, Cu_2Se , Bi_2O_3 and $CuSe-BiO$ were calculated and displayed in **Fig. SR23a**. Clearly, the BiCuSeO still exhibited the maximum and even greatly increased current density at all potentials, suggesting a large increase of intrinsic activities refer to Bi_2O_3 and Cu_2Se . The above results and detailed analysis are added in the revised manuscript as **Fig. S9**, **Fig. S10** and **Table S3**.

Fig. SR22 CV curves with different scan rates. **a** BiCuSeO, **b** Cu₂Se, **c** Bi₂O₃. **d** CuSe-BiO

Fig. SR23 **a** C_{dl} comparisons by dividing the double layer charging current differences with the scan rates. **b** ECSA-normalized formate current densities

Table SR7. ECSA calculations results

Smamples	Cdl (mF cm^{-2})	ECSA
BiCuSeO	5.66	94.3
Cu ₂ Se	5.59	93.2
Bi ₂ O ₃	4.91	81.8
CuSe-BiO	4.48	74.7

At the end, we acknowledge again for all the recognition and constructive comments and suggestions from all the reviewers. We are sure that the quality of this work has been greatly improved.

REVIEWER COMMENTS

Reviewer #1 (Remarks to the Author):

The revised work is comprehensive, and the paper seems to be nicely written. It can be proceeded to further step.

Reviewer #2 (Remarks to the Author):

I agree with the revision made the authors. My concerns are fully addressed. I would like to recommend this manuscript as is.

Reviewer #3 (Remarks to the Author):

The authors have addressed most of my concerns. However, I have two remaining suggestions for the authors to consider before official acceptance of this work in Nature Communications:

1. Based on my previous Comment 2, the authors combine DFT and experiments to show the stability of BiCuSeO lattice during CO₂RR, but only in a qualitative manner. I suggest the authors quantify (e.g. inductively coupled plasma mass spectrometry (ICP-MS)) the amount of Se loss and refine the DFT model accordingly to show the integrity of the BiCuSeO superlattice EVEN with significant Se losses.

2. Based on my previous Comment 10, the authors cited a review article (Adv. Mater., 32, 2001848 (2020)) to show a thermodynamic potential (E_0) for CO₂-to-formate conversion is -0.21V, which I think is not accurate. As detailed in a previous report (Fig. S9 in Nano Energy 31, 270-277 (2017)), the E_0 of formate is clearly pH-dependent. In the case of 0.5 M KHCO₃, $E_0 = -0.09$ V vs. RHE, resulting in an overpotential of 310 mV, instead of 190 mV claimed by the authors, for formate production in this work.

With those addressed, the article should be accepted.

Response Letter

Overall response: We thank the reviewers and editor for the thoughtful review and critical comments about our manuscript. We welcome the opportunity to address and clarify the issues raised in the reviewers' report, and believe that the additional revisions carried out to address the reviewers' comments substantially strengthen our revised manuscript. The additional data and revisions in the revised manuscript have been highlighted with a yellow background.

Response to Reviewer 1

Comment: The revised work is **comprehensive**, and the paper seems to be **nicely written**. It can be proceeded to further step.

Response: We greatly appreciate the approval of our work.

Response to Reviewer 2

Comment: I agree with the revision made the authors. My concerns are fully addressed. I would like to recommend this manuscript as is.

Response: We greatly appreciate the approval of our work.

Response to Reviewer 3

Overall comment from Reviewer 3's: The authors have addressed most of my concerns. However, I have two remaining suggestions for the authors to consider before official acceptance of this work in Nature Communications.

Response: We greatly appreciate the approval of our work. In the revised version, the amount of Se loss was quantified by ICP-MS and XAS, and the integrity of the BiCuSeO superlattice was calculated using the ab initio molecular dynamics (AIMD) simulation to address the concerns of the reviewer. The additional data and revisions were marked by yellow highlight background.

Comment 3-1: Based on my previous Comment 2, the authors combine DFT and experiments to show the stability of BiCuSeO lattice during CO₂RR, but only in a qualitative manner. I suggest the authors quantify (e.g. inductively coupled plasma mass spectrometry (ICP-MS)) the amount of Se loss and refine the DFT model accordingly to show the integrity of the BiCuSeO superlattice EVEN with significant Se losses.

Response: Thanks for the reviewer's significant comment. This is a very important issue. Following your suggestion, the amount of Se loss is quantified using ICP-MS test and XAS spectra. ICP-MS test shows that ~11 atom% Se atoms are retained in the BiCuSeO sample after electroreduction CO₂ reaction (BiCuSeO_R) in 0.5 M KHCO₃ electrolyte. Moreover, Cu-Se coordination number in the BiCuSeO_R sample is ~0.7 according to EXAFS fitting (**Table S8**). In compared with the pristine BiCuSeO (Cu-Se coordination number is 4), it can be calculated that ~17.5 atom% Se atoms are reserved in BiCuSeO_R, which is consistent with ICP-MS result. In order to gain a theoretical insight into the stability of structures, The ab initio molecular dynamics (AIMD) simulation was carried out, which has been widely used to investigate the thermodynamic and dynamic stability of the solid materials, especially for 2D materials (*Proc. Natl Acad. Sci. USA* 102, 6654-6659 (2005); *J. Phys. Chem. Lett.* 12, 12230-12234 (2021)). The AIMD simulation is a method that perform the molecular dynamics by calculating electronic structures "on the fly" while retaining a first-principles-based description of their interactions. In our AIMD simulation, the electronic Schrodinger equation is approximated using Kohn-Sham density functional theory (DFT). As the reviewer commented, the corresponding theoretical model is constructed according to the above quantitative analysis. Accordingly, AIMD is employed to investigate the integrity of the BiCuSeO superlattice with ~87.5 atom% Se losses (**Fig. SR1 c, d**, only with ~12.5 atom% Se

atoms are retained) under 300 K. The pristine BiCuSeO is also simulated for convenient comparison. (**Fig. SR1 a, b**). The whole AIMD simulation is performed within the NVT ensemble and the whole process is lasted 5 ps with a time step of 1.0 fs. The calculated results show the stable energy (E_0) and no obvious structural collapse in both the BiCuSeO_R with 12.5 atom% and the pristine BiCuSeO, indicating a good integrity of the BiCuSeO superlattice with even significant Se losses. Moreover, SEM images (**Supplementary Fig. S11**), TEM images (**Fig. 5f, Supplementary Fig. S14d**), HRTEM image (**Supplementary Fig. S14e**) and SAED pattern (**Supplementary Fig. S14f**) clearly show that the structural framework of superlattice tends to mainly retained. As proposed in **Fig. 5a**, in the CO₂RR process, most Se atoms have escaped from the lattice, at the same time, some O atoms enter and substitute the Se atoms with weaker electronegativity and finally form Cu-O bonds. The eventually derived BiCuSeO_R after the CO₂RR inherits the superlattice structure.

Fig. SR1 Energy variation and structural evolution with AIMD simulation. **a, b** the pristine BiCuSeO. **c, d** BiCuSeO_R with 12.5 atom% Se.

Fig. S11 **a** Pristine BiCuSeO and **b** BiCuSeO_R samples collected after 10 h electrochemical measurements for CO₂RR.

Fig. S14 Structural characterizations of BiCuSeO_R samples collected after CO₂RR. **a**, **b** XRD pattern. **c** Raman spectrum. **d** TEM image. **e** HRTEM image. **f** SAED pattern.

Comment 3-2: Based on my previous Comment 10, the authors cited a review article (Adv. Mater., 32, 2001848 (2020)) to show a thermodynamic potential (E^0) for CO₂-to-formate conversion is -0.21V, which I think is not accurate. As detailed in a previous report (Fig. S9 in Nano Energy 31, 270-277 (2017)), the E^0 of formate is clearly pH-dependent. In the case of 0.5 M KHCO₃, $E_0 = -0.09$ V vs. RHE, resulting in an overpotential of 310 mV, instead of 190

mV claimed by the authors, for formate production in this work. With those addressed, the article should be accepted.

Response: Thanks for the reviewer's insightful comment. We agree with the reviewer's point of view that "the potential of formate is pH-dependent". Similarly, it is well known that RHE (reversible hydrogen electrode) is linearly related to the pH of the electrolyte. And, the potential of RHE (E_{RHE}) can be calculated according to the formula $E_{\text{RHE}} = -0.0591\text{pH}$ (25 °C, 1atm). For better comparisons, the test potentials (E) in reaction of electroreduction CO_2 into formate are usually converted into those refer to RHE according to the formula

$$E = E^0 + E_{\text{RHE}} + \frac{2.3RT}{nF} \log \frac{a_{\text{HCOOH}}}{a_{\text{HCOO}^-}}$$
, where R is constant ($R = 8.314 \text{ J mol}^{-1} \text{ K}^{-1}$), T is temperature, n is electron transfer number, F is Faraday constant. If the reaction

$\text{HCOOH} \leftrightarrow \text{HCOO}^- + \text{H}^+$ is not taken in account, CO_2 is electroreduced into formate according to the equation $\text{CO}_2 + 2\text{H}^+ + 2\text{e} \leftrightarrow \text{HCOOH}$. And, the formula

$$E = E^0 + E_{\text{RHE}} + \frac{2.3RT}{nF} \log \frac{a_{\text{HCOOH}}}{a_{\text{HCOO}^-}}$$
 can be simplified into $E = E^0 + E_{\text{RHE}}$. Obviously, the

potential of formate relative to RHE (E^0) is pH independent. Moreover, many reports and calculation results show that the Gibbs energy for CO_2 reduction reaction into formic acid (ΔG^0 , 25 °C, 1 atm) is approximately 0.4 eV, and the electrode potential (E^0) of approximately -0.21 V vs. RHE can be gained according to the formula $\Delta G^0 = nFE^0$. Even more, the potential approximately -0.21 V (RHE) for formate production from CO_2RR reaction in 0.5 M KHCO_3 aqueous electrolyte under 1.0 atm, 25°C is use in the above literatures (such as *Nat. Nanotech.* 16, 1386-1393, (2021); *Nat. Energy* 4, 776-785 (2019); *Nat. Commun.* 11, 1088 (2020); *Nat. Commun.* 10, 892 (2019); *Nat. Commun.* 9, 1320 (2018); *Angew. Chem. Int. Ed.* 58, 14197-14201 (2019); *Angew. Chem. Int. Ed.* 58, 13828-13833 (2019); *Angew. Chem. Int. Ed.* 58, 8499-8503 (2019); *Angew. Chem. Int. Ed.* 57, 12790-12794 (2018); *Adv. Mater.* 32, 2001848 (2020)). Therefore, the overpotential for formate generation is calculated based on $E^0=0.21 \text{ V}$ (RHE) in our manuscript.

REVIEWERS' COMMENTS

Reviewer #3 (Remarks to the Author):

I am fully satisfied with the authors response and final version, and I recommend publication.